# AP-1 and TGFß cooperativity drives non-canonical Hedgehog signaling in resistant basal cell carcinoma

Catherine D. Yao[1], Daniel Haensel [1], Sadhana Gaddam[1], Tiffany Patel[1], Scott X. Atwood [1,5], Kavita Y. Sarin [1], Ramon J. Whitson[1,2], Siegen McKellar [1,3], Gautam Shankar[1,4], Sumaira Aasi[1], Kerri Rieger [1] & Anthony E. Oro [1✉]

Tumor heterogeneity and lack of knowledge about resistant cell states remain a barrier to targeted cancer therapies. Basal cell carcinomas (BCCs) depend on Hedgehog (Hh)/Gli signaling, but can develop mechanisms of Smoothened (SMO) inhibitor resistance. We previously identified a nuclear myocardin-related transcription factor (nMRTF) resistance pathway that amplifies noncanonical Gli1 activity, but characteristics and drivers of the nMRTF cell state remain unknown. Here, we use single cell RNA-sequencing of patient tumors to identify three prognostic surface markers (LYPD3, TACSTD2, and LY6D) which correlate with nMRTF and resistance to SMO inhibitors. The nMRTF cell state resembles transit-amplifying cells of the hair follicle matrix, with AP-1 and TGFß cooperativity driving nMRTF activation. JNK/AP-1 signaling commissions chromatin accessibility and Smad3 DNA binding leading to a transcriptional program of RhoGEFs that facilitate nMRTF activity. Importantly, small molecule AP-1 inhibitors selectively target LYPD3+/TACSTD2+/LY6D+ nMRTF human BCCs ex vivo, opening an avenue for improving combinatorial therapies.

[1] Program in Epithelial Biology, Stanford University School of Medicine, 269 Campus Drive, Stanford, CA 94305, USA. [2] Genomics Institute of the Novartis Research Foundation, La Jolla, CA, USA. [3] University of Washington School of Medicine, Seattle, WA 98195, USA. [4] Johns Hopkins School of Medicine, Baltimore, MD 21287, USA. [5] Present address: Department of Developmental and Cell Biology, University of California, Irvine, USA. ✉email: oro@stanford.edu

While great strides have been made in developing oncogene-targeted cancer therapies, tumor heterogeneity continues to limit the efficacy of monotherapies. An ongoing challenge in the field is the ability to identify and define the resistant cell states within tumors and the driving factors in order to apply precision combination therapies. While a myriad of cell types could exist, single-cell technologies support the accumulation of a finite number of cell types in each tumor that resemble normal developmental cell types, albeit with aberrant differentiation programs[1,2].

Basal cell carcinoma (BCC) of the skin is the most common cancer in the United States, and serves as an excellent model for the study of tumor evolution and heterogeneity, due to its repeated UV exposure, high mutation burden[3–6], and convenience in obtaining clinical samples[7]. BCC is singularly driven by over-activation of the essential Hedgehog (Hh) pathway, making it a promising candidate for targeted therapies. However, disappointing initial response rates of only 40% were found with SMO inhibitor therapies such as FDA-approved vismodegib, with a recurrence rate of 20% during the first year[8–10]. While initial studies focused on identifying Hh pathway mutations that conferred acquired resistance[3], our subsequent studies revealed intrinsically resistant cell states driven by non-canonical Hh signaling in naive patient BCCs as another significant contributor to targeted therapy failure.

We previously identified a common resistant cell state where RhoA increases actin polymerization and nuclear translocation of myocardin-related transcription factor (nMRTF)[11]. nMRTF binds to serum response factor (SRF) and acts as a positive transcriptional cofactor for Gli at Hh target gene sites[12]. Interrogation of the frequency of nMRTF within clinical BCC samples revealed that up to 60% of naive BCCs display varying levels of nMRTF and are intrinsically resistant to SMO inhibitors ex vivo, despite no prior exposure to treatment[12]. This observation suggests that the nMRTF phenotype exists as a distinct cell state in heterogenous BCC. However, the origin, cellular definition, and driving transcription factors (TFs) that characterize this resistant sub-population remain unknown.

While BCC tumors depend on Hh/Gli signaling for proliferation and inhibition of differentiation, the pathway exists at a signaling nexus that interacts with other pathways to integrate environmental and tumor state-dependent signals. The co-incidence of multiple signals simultaneously is necessary to exert highly specific, dynamic chromatin changes. For example, it is well known that TGFß/Smad TFs play a complex role in cancer progression, acting as either a tumor suppressor or promoter depending on the context of their activation and which TFs are coactivated[13]. Indeed, there is precedence for dysregulated TGFß signaling in BCC[14], and our group has recently uncovered a role for TGFß in governing BCC invasiveness[15]. Similarly, while AP-1 and its various Jun and Fos family members are well-described regulators of keratinocyte proliferation and homeostasis[16,17], their function in BCC tumorigenesis remains unstudied.

Here we harness single-cell RNA sequencing (scRNA-seq) of patient tumors to identify surface markers that reliably enrich for nMRTF cells and predict responsiveness to inhibitor treatments. We further show that the unique chromatin accessibility and transcriptional programs of the nMRTF cell type closely resemble the differentiation state of suprabasal cells of the hair follicle (HF) matrix and are driven by JNK/Jun and TGFB/Smad3 signaling, highlighting these pathways for combination therapies in resistant BCC.

BCCs confers Hh target gene transcription and persistent growth in the presence of SMO inhibitors (SMOi) including vismodegib[12]. However, the key driving factors that lead to active nMRTF remain unclear. To understand how MRTF interacts with the Hh pathway in the context of normal skin homeostasis, we examined MRTF cellular localization in mouse skin. Intriguingly, while BCCs are thought to derive from the bulge and outer root sheath (ORS)[18], we found that nuclear MRTF is restricted specifically to the matrix transit-amplifying cells (TACs) of the HF and dermal papilla (DP) (Fig. 1a, Supplementary Fig. 1a), with other sub-domains of the skin showing predominantly cytoplasmic localization. While both the ORS and TACs respond to Shh signaling, TACs are committed HF progenitors that undergo Shh-dependent expansion during anagen. Notably, the HF matrix contains cells that express the highest levels of Hh signaling[19], and they are known to be highly responsive to TGFß and Wnt secreted from the DP[20,21] (Fig. 1a, Supplementary Fig. 1a).

To further characterize the differentiation state of nMRTF cells, we interrogated MRTF cellular localization in clinical samples of benign skin tumors with known cellular origins. Strong and uniform nuclear MRTF was observed in HF matrix-derived pilomatricomas (PMX), while epithelial inclusion cysts (EIC) derived from interfollicular epidermis (IFE) as well as trichilemmal/pilar cysts (PC) derived from ORS displayed cytoplasmic MRTF localization[22,23] (Fig. 1b). Furthermore, we found that out of 1201 MRTF target genes identified in resistant BCCs (Supplementary Data 1), 65% of those genes are upregulated in basal and/or suprabasal TACs[24]. These observations reinforce the relation of MRTF activity with the hair matrix lineage.

In parallel, we took a global approach to analyze the chromatin landscape relationships between BCC and normal skin by comparing the Assay for Transposase Accessible Chromatin (ATAC-seq) profiles of resistant and sensitive BCCs with those of various epithelial cell types (Fig. 1c, Supplementary Fig. 1b, Supplementary Table 1). We compared open chromatin of murine nMRTF BCC cell line ASZ001[25], previously selected to be highly resistant to SMOi-induced changes in cell growth (res-BCC)[26], with other published ATAC-seq datasets from sensitive BCC generated from *K14-creER; Ptch1^{fl/fl};Tp53^{f/f}* mice (sens-BCC)[27], basal and suprabasal TACs (basal TAC and suprabasal TAC), bulge HF stem cells, hair germ (HG)[24], and interfollicular epidermal stem cells (IFE)[28]. Principal component analysis (PCA) representation of these relationships reveals that the chromatin accessibility profiles of nMRTF resistant BCC cells cluster most closely with those of suprabasal and basal TAC, while sensitive BCC clusters with HG and bulge stem cells (Fig. 1c and Supplementary Fig. 1b).

We also compared nMRTF cells to residual BCC cells (resid-BCC), reversible Wnt-dependent, IFE-like tumor cells identified in a murine BCC model after vismodegib treatment[27]. We see that multiple resistant nMRTF BCC cell lines (ASZ and BSZ[25]) have quite disparate chromatin accessibility profiles from sensitive BCC as well as residual BCC (Supplementary Fig. 1d). Furthermore, the genes with increased chromatin accessibility in resistant nMRTF BCC comprise a uniquely pro-proliferative, pro-migratory program, suggesting they represent a distinct cell fate (Supplementary Fig. 1e). Significant phenotypic differences also exist between these BCC cell types, as nMRTF cells maintain high Hh signaling and proliferation rates when treated with vismodegib[12] (see below), while residual tumor cells resume growth only when treatment is discontinued[27]. We conclude that SMOi-resistant nMRTF BCCs possess a distinct cellular state within naive tumors most closely resembling HF matrix TACs.

The observation that MRTF-SRF-Gli chromatin occupancy confers SMO-independent enhancement of Gli activity suggests the intriguing hypothesis that nMRTF functions in the hair to extend hair matrix proliferation at a distance from the Shh signal

## Results
**nMRTF BCC chromatin accessibility resembles HF matrix.** We have previously shown that active nuclear MRTF (nMRTF) in

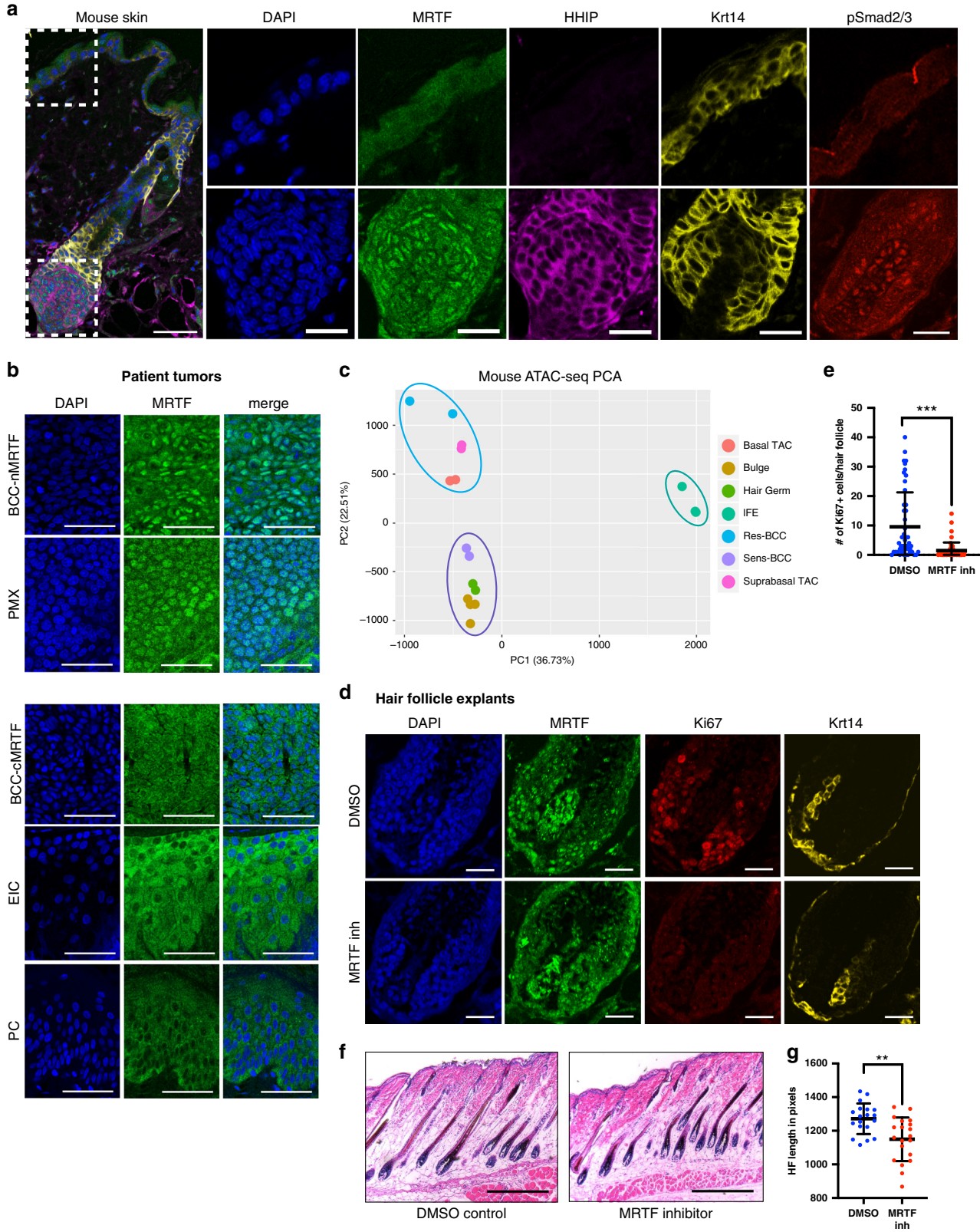

localized to the lateral disc[29]. Indeed, in mouse anagen HFs, proliferating matrix cells extend to the line of Auber on average 4–5 cell divisions and cease as nMRTF becomes cytoplasmic (Fig. 1a, d). We tested this hypothesis by treating mouse HF explants with CCG-1423, an MRTF-inhibitor (MRTFi), and found markedly reduced proliferation in the matrix (Fig. 1d, e) as measured by Ki67 staining, and a concomitant decreased total HF length (Fig. 1f, g). Altogether, we conclude that MRTF nuclear localization is linked to a distinct cell state of differentiation, allowing it to extend the cellular response to Shh signaling.

**Fig. 1 nMRTF BCC chromatin accessibility resembles hair follicle matrix. a** IF representative images of MRTF cellular localization in healthy mouse skin, in comparison to localization of HHIP (Hh-responsive matrix transit-amplifying cells), Keratin-14 (interfollicular epidermis and outer root sheath), and phospho-Smad2/3 (TGFß-responsive dermal papilla and matrix transit-amplifying cells). Inset areas (IFE top row, HF matrix bottom row) denoted by white dotted boxes. Single-color images are all derived from the same slide except for pSmad2/3, due to limitations of antibody co-staining. Images are representative of $n > 50$ hair follicles examined. Scale bar at low power = 50 μm, inset = 25 μm. **b** IF images of MRTF localization in clinical biopsies of basal cell carcinoma with nuclear or cytoplasmic MRTF (BCC-nMRTF or -cMRTF), pilomatricomas (PMX), epithelial inclusion cysts (EIC), and pilar cysts (PC). Images are representative of $n = 3$ patient tumors per subtype. Scale bar = 50 μm. **c** Principal component analysis (PCA) plot showing the relationship of ATAC-seq profiles between resistant BCC cell line ASZ001 (res-BCC), sensitive murine BCC (sens-BCC), basal transit-amplifying cells of hair follicle (basal TAC), bulge hair follicle stem cells (Bulge), hair germ (HG), interfollicular epidermal stem cells (IFE), and suprabasal transit-amplifying cells of the hair follicle (suprabasal TAC). See Supplementary Table 1 for data sources. **d** IF images of MRTF, Ki67, and Krt14 in mouse hair follicle skin explants treated with DMSO vehicle or 150 μM MRTF[i] CCG-1423 for 72 h. Scale bar = 20 μm. **e** Quantification of Ki67+ cells per hair follicle in (**d**). $n = 4$ pairs of explants from three mice. ***$p < 0.0001$. **f** Representative H&E staining of DMSO or CCG-1423 treated mouse skin explants. Scale bar = 100 μm. **g** Quantification of hair follicle lengths in (**f**). Each point represents the length of one HF averaged over up to three separate measurements. **$p = 0.0017$. All error bars represent mean +/− SD. $p$-values were calculated using unpaired, two-tailed Student's $t$-test.

**LYPD3/TACSTD2/LY6D mark the nMRTF population in patient BCCs.** We have shown previously that naive human BCCs, such as those excised through standard Mohs surgical techniques, contain a heterogenous number of cells displaying active nMRTF, which is predictive of the effectiveness of MRTF[i] versus SMO[i] for BCC treatments[12]. Therefore, we took advantage of the naturally occurring heterogeneity and conducted scRNA-seq on four naive human BCC tumors to identify and segregate the MRTF-active vs. inactive subpopulations for further study (Supplementary Fig. 2a, b). From the 45,656 total cells analyzed, the epithelial tumor cells were segregated from fibroblast, endothelial, and immune populations through Krt14 expression (Supplementary Fig. 2c–g). Samples were batch-corrected using canonical correlation analysis (CCA) before dimensional reduction and visualization of clusters via tSNE (Fig. 2a).

Cells with the highest MRTF activity were identified through an enrichment score based on the expression of a signature list of MRTF target genes (Fig. 2b). The list is derived from the intersection of SRF target genes identified by ChIP-seq, and MRTF-dependent genes identified by RNA-seq in resistant BCC cells treated with MRTF inhibitor CCG-1423[12] (Supplementary Fig. 2h, Supplementary Data 1). Clusters were then ranked based on their mean MRTF signature enrichment score, and the 12 clusters were separated into four groups: high MRTF, med-high, med-low, and low MRTF (Fig. 2c). Interestingly, the four tumors were not equally distributed among the clusters, as evidenced by the significant chi-squared statistic. The high MRTF clusters contained higher percentages of two of the four tumors (BCC8 and BCC3), consistent with our previous observation that at least 50% of naive clinical BCC samples contain nMRTF[12] (Supplementary Fig. 2i).

We then determined the most highly upregulated genes in the high MRTF group vs. low MRTF group (Fig. 2d). To separate MRTF-active BCC cells by fluorescence-activated cell sorting (FACS), we selected target genes of MRTF whose protein products are expressed on the cell surface. The top three most highly enriched surface markers in the MRTF high group were LYPD3, TACSTD2 (also known as TROP-2), and LY6D (Fig. 2e, Supplementary Fig. 2j–l). To validate the specificity of these surface markers, we immunostained human and mouse BCC tumors along with MRTF. We found the protein levels of all three markers was significantly higher in tumor regions with nMRTF (Fig. 2f, g). Interestingly, the surface markers are also expressed at high levels in the HF matrix in normal mouse skin, correlating with nuclear MRTF (Fig. 2f). We also treated naive patient BCC explants with MRTF[i] CCG-1423 and observed significantly reduced protein levels of all three surface markers, further confirming that expression of these markers is dependent on MRTF activity (Fig. 2h). To correlate the expression of the surface

markers with functional resistance, we treated naive patient BCC explants with SMO[i] vismodegib, and found that protein expression of all three markers is significantly correlated with maintenance of Gli1 mRNA levels, indicating diminished response to SMO inhibition ex vivo (Fig. 2i). These data confirm that the three surface proteins LYPD3, TACSTD2, and LY6D are reliable indicators of nuclear MRTF activity as well as prognostic markers of SMO[i] resistance in human BCC cells.

To further investigate the chromatin landscape and transcriptomic differences specific to resistant nMRTF BCC, we sorted fresh naive clinical BCC samples first based on epithelial ITGA6 expression, then on these three surface markers (Fig. 2j, Supplementary Fig. 2m). Interestingly, we see significant co-expression of LYPD3 and TACSTD2, while LY6D shows much scarcer expression levels, consistent with our scRNA-seq results (Fig. 2e, Supplementary Fig. 2j–l). For this reason, we sorted ITGA6+ LYPD3+ TROP2+ LY6D+/− cells as surface marker positive (SM+) population and ITGA6+ LYPD3− TROP2− LY6D− cells as surface marker negative (SM−) and compared SM+ and SM− RNA-seq and ATAC-seq profiles (Supplementary Fig. 2p, q). Importantly, the expression of the MRTF signature gene list was significantly enriched by GSEA analysis in the SM+ population, further validating the three surface proteins as reliable markers of MRTF activity (Fig. 2k, l). As additional support of the specificity of these surface markers for resistant nMRTF BCC, we examined their expression in Gorlin syndrome patient BCCs, which are caused by inherited loss of *ptch1* and respond to vismodegib with almost no resistance[30], as well as being consistently MRTF-inactive[12]. Indeed, Gorlin's syndrome BCC tumor cells lack expression of the three surface markers (Supplementary Fig. 2n). Chromatin accessibility profiles of Gorlin BCCs are more similar to sorted SM- than SM+ naive patient BCCs by PCA (Supplementary Fig. 2o), providing further support that the SM+ surface phenotype can prospectively enrich for BCC cells possessing the distinct resistant nMRTF cell fate, allowing further characterization from heterogeneous BCC populations.

**BCC resistance requires coincident AP-1 and TGFß signaling.** We used BETA analysis[31] to integrate transcriptomic and chromatin accessibility data in order to identify differentially regulated genes and TF motifs between SM+ and SM− patient BCC cells (Supplementary Fig. 3a). Interestingly, the top Gene Ontology (GO) Biological Process terms[32] enriched in the SM+ cells included terms associated with epidermal development and keratinocyte differentiation, underscoring the idea that resistant nMRTF BCC resemble a more differentiated cell state than sensitive BCC (Fig. 3a). To find driver TF pathways, we looked at enriched TF motifs as well as target gene expression from Chip-X Enrichment Analysis (ChEA)[33] (Fig. 3b, c). Notably, AP-1, Smad2/3, and p63

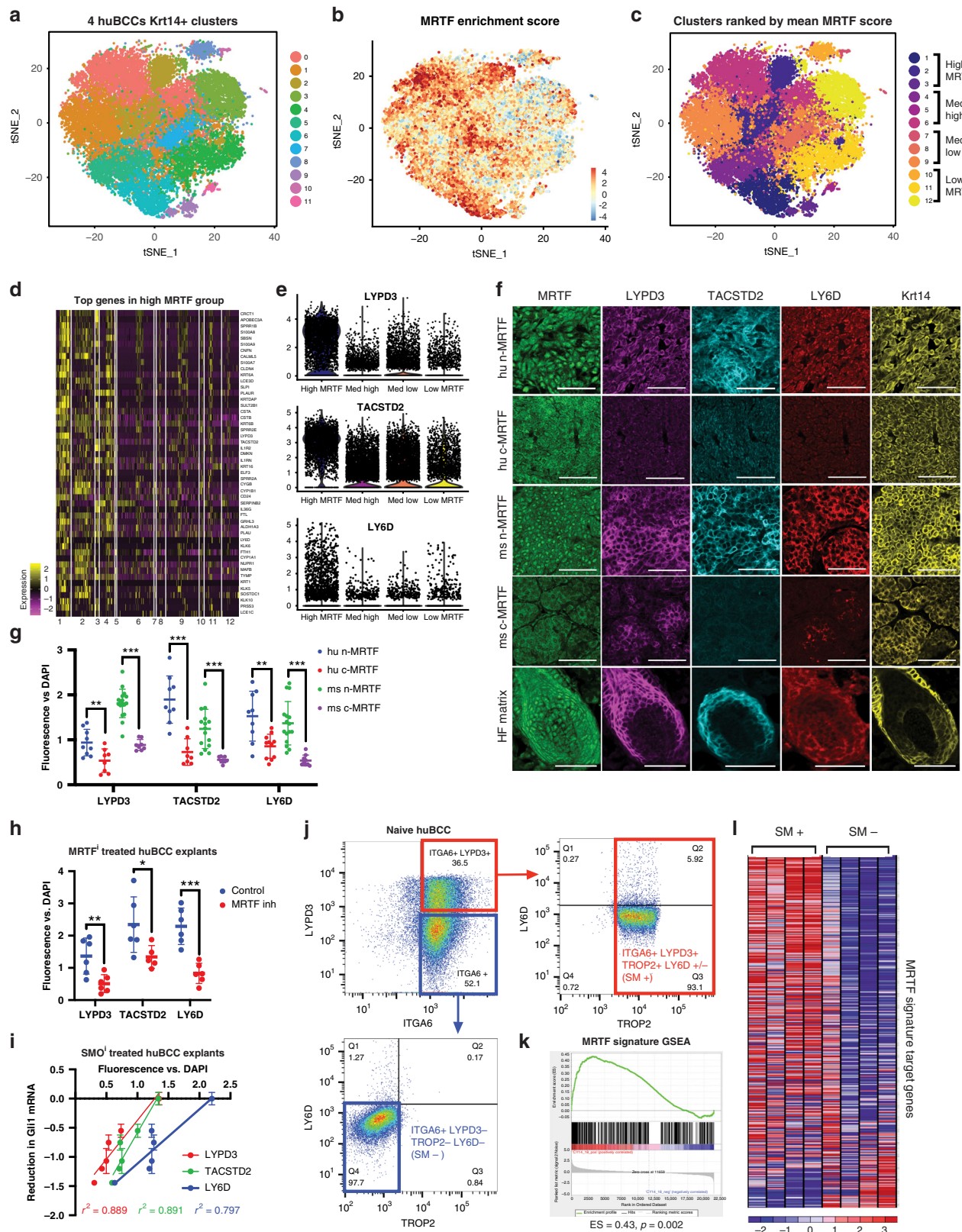

were among the top TFs acting in SM+ cells. As p63 is a known master regulator for epidermal development, we focused on AP-1 and Smad as potential activators of MRTF and non-canonical Hh signaling.

First, we assessed the necessity of TGFß and AP-1 signaling in resistant BCC through genetic and pharmacologic perturbation.

Treatment of BCC cells with various AP-1 small-molecule inhibitors (T5224[34] and SR11302[35]) resulted in a dose-dependent decrease in expression of *Gli1* measured by mRNA and protein, and cell viability (Fig. 3d, e and Supplementary Fig. 3b, c). We also observed elevated levels of phosphorylated JNK in active nMRTF BCCs in comparison to inactive cMRTF BCC (Supplementary

**Fig. 2 LYPD3/TACSTD2/LY6D mark the nMRTF subpopulation in patient BCCs. a** tSNE plot of unbiased clustering of tumor epithelia from 4 naive human BCC tumor scRNA-Seq datasets post multi-CCA alignment, filtered for positive Krt14 expression. **b** MRTF signature enrichment score (generated from genes in Supplementary Data 1) overlaid on tSNE clusters from (**a**). c Clusters ranked by mean MRTF signature enrichment score, then grouped as follows: clusters 1–3 = high MRTF, clusters 4–6 = med high, clusters 7–9 = med low, clusters 10–12 = low MRTF. **d** Heatmap of top 50 genes most enriched in high MRTF group (clusters 1–3) vs. low MRTF group (clusters 10–12). **e** Violin plots of gene expression levels per group for surface markers *LYPD3*, *TACSTD2*, and *LY6D*. **f** Representative IF images of surface marker expression in human and mouse MRTF-nuclear vs. MRTF-cytoplasmic BCC regions or HF matrix of normal mouse skin. scale bar = 50 µm. **g** Quantification of LYPD3, TACSTD2, and LY6D fluorescence intensity normalized to DAPI. Each point represents mean pixel intensity, normalized to mean DAPI intensity, averaged over at least three 100 × 100 µm microscopy fields. **p < 0.01, ***p < 0.001. **h** Quantification of LYPD3, TACSTD2, and LY6D fluorescence intensity normalized to DAPI of naive patient BCC explants treated with 10 µM MRTF[i] CCG-1423 for 24 h. Each point represents mean pixel intensity, normalized to mean DAPI intensity, averaged over at least three 100 × 100 µm microscopy fields. *p = 0.0384, **p = 0.0072, ***p = 0.0002. All error bars represent mean +/− SD. p-values calculated using unpaired, two-tailed Student's t-test. **i** Relative reduction in Gli1 mRNA levels as measured by qRT-PCR vs. quantified fluorescence intensity of surface markers normalized to DAPI in naive human BCC (huBCC) explants treated with 1 µM vismodegib for 24 h. Respective linear regression $r^2$ values shown in matching colors. **j** Representative FACS plots showing distributions of ItgA6 (CD49f), LYPD3, TACSTD2 (TROP2), and LYPD3 protein expression in naive human BCC tumors. Final sorted populations taken for further analysis outlined in red and blue. **k** Gene Set Enrichment Analysis (GSEA) plot of MRTF signature gene list comparing RNA-seq of sorted surface marker positive (SM+) vs. negative (SM−) BCC cells. n = 4 replicates from two tumors. ES = enrichment score, p = nominal p-value[64]. **l** Gene expression heatmap from GSEA in (**k**).

Fig. 3g, h), and treatment of resistant BCC cells with inhibitors of JNK (SP600125[36] and JNK-IN-8[37]) reduce *Gli1* expression and cell viability (Supplementary Fig. 3d). By contrast, inhibitors of p38 or MEK fail to demonstrate similar inhibition[12], suggesting that AP1/JNK signaling mediates resistance. In parallel, a small-molecule inhibitor of TGFß signaling through ALK5 (SB435142[38]) leads to a dose-dependent decrease in phosphorylated Smad3 levels (Supplementary Fig. 3e), as well as *Gli1* expression and cell viability (Fig. 3d-e). Both AP-1 inhibitor and ALK5 inhibitor were preferentially toxic to BCC cells, as they affected cell viability of multiple BCC cell lines[25] to a significantly higher degree than noncancerous cell lines (Supplementary Fig. 3i), providing key pre-clinical data for a drug therapeutic window. Notably, the combination of AP-1 and ALK5 inhibitors on resistant BCC cells did not result in any additional effect on *Gli1* expression or cell viability, suggesting that these pathways may be redundant or acting upstream of a shared pathway (Supplementary Fig. 3f).

We next wanted to determine the specific AP-1 and TGFß family members that operate in resistant BCCs. Although AP-1 consists of a family of Jun/Fos dimers all capable of binding to the TGA(C/G)TCA consensus sequence, different homo or heterodimer pairs can have drastically different transcriptional outputs[39]. *c-Jun, JunB*, and *JunD* are all expressed robustly in BCC cells, while *FosL2* is the only Fos family member expressed at significant levels (Supplementary Fig. 3k). Consistent with this observation, siRNA knockdowns of *c-Jun, JunB, JunD*, or *FosL2* in BCC cells resulted in significant decrease in *Gli1* expression, with *JunD* knockdown having the biggest effect, while knockdown of *Fos* or *FosB* did not change *Gli1* expression (Fig. 3f, Supplementary Fig. 3l). In parallel, siRNA knockdown of TGFß family members *TGFß1, TGFß3, Smad3*, and *Alk5* decreased Smad2/3 phosphorylation (Supplementary Fig. 3m) and reduced expression of Gli1 (Fig. 3f, Supplementary Fig. 3l, m).

nMRTF activity depends on RhoA activation[12], so we next examined whether AP-1 and TGFß cooperate to induce Rho signaling. Using a Rho G-LISA assay to specifically measure levels of active GTP-bound RhoA, we found that exogenous TGFß3 ligand was sufficient to activate Rho signaling, which could be attenuated by the addition of ALK5 or AP-1 inhibitor (Fig. 3g). In addition, we found that inhibition of TGFß or AP-1 signaling disrupted both actin polymerization and MRTF nuclear localization (Fig. 3h, i). Importantly, while TGFß signaling regulates Rho activity in both Smad-dependent and independent pathways[40], knockdown of *Smad3* expression is sufficient to inhibit MRTF nuclear localization (Fig. 4g) and *Gli1* expression (Fig. 3f). This

suggests that in resistant BCCs, TGFß acts in a Smad3-dependent manner to promote non-canonical Hh signaling.

If AP-1 and/or TGFß promote a non-canonical resistance mechanism to drive Hh signaling, then inhibiting these TF pathways should increase the sensitivity of cells to canonical SMO[i] vismodegib. Indeed, we see that combining SMO[i] treatment with AP-1 inhibition or ALK5 inhibition in resistant BCC cells leads to increased levels of cell death than with SMO[i] alone (Fig. 3j). In contrast, neither AP-1 inhibition nor ALK5 inhibition have any additive effect on cell viability when combined with MRTF[i] CCG-1423 (Supplementary Fig. 3j), suggesting that these pathways work upstream of MRTF. These initial experiments connect JNK-mediated signaling via Jun/FosL2 and TGFß signaling via Smad3 to maintain activation of Rho and MRTF, leading to non-canonical Hedgehog signaling in resistant BCC cells.

**AP-1 and Smad3 induce transcription of Rho GEFs.** To identify how AP-1 and Smad3 transcriptional targets activate Rho, we performed RNA-seq on resistant BCC cells treated with ALK5 or AP-1 inhibitors. Intriguingly, the top Molecular Function GO terms of genes dependent on both pathways included guanyl-nucleotide exchange factor (GEF) activity, proteins that directly activate Rho family GTPases (Fig. 4a–c, Supplementary Data 2). Previous studies have shown that TGFß regulates Rho GTPases through transcription of GEFs in contexts such as epithelial-mesenchymal transition, making these promising candidate genes[41,42]. Targeted siRNA knockdown of the differentially expressed RhoGEFs identified the set of GEFs that work together to maintain *Gli1* expression (Fig. 4d, Supplementary Fig. 4a, b). Although maximal Rho activity required several exchange factors, Arhgef17, also known as Tumor Endothelial Marker 4, was the most significantly enriched at the protein level in mouse resistant BCC tumors with nuclear MRTF, generated from transgenic mouse model *Ptch1*[+/−];*K14-creER;p53*[fl/fl][43] (Fig. 4e, f). Arhgef17 facilitates GDP/GTP exchange for RhoA and is required for cell-cell adhesion[44]. Knockdown of *Arhgef17* as well as related GEF *Arhgef18* using fluorescently labeled siRNAs phenocopied *Smad3, JunD*, and other AP-1 subunit knockdowns in abrogating MRTF nuclear localization (Fig. 4g, h and Supplementary Fig. 4c).

These findings suggest coincident TGFß and AP-1 signaling confers resistance by increasing transcription of Rho GEFs such as Arhgef17, which in turn activate RhoA and subsequent actin polymerization, leading to nuclear localization of MRTF and SRF, which as act non-canonical cofactors for Gli1 (Fig. 4i)[12]. In order to confirm this resistance pathway, we conducted a series of

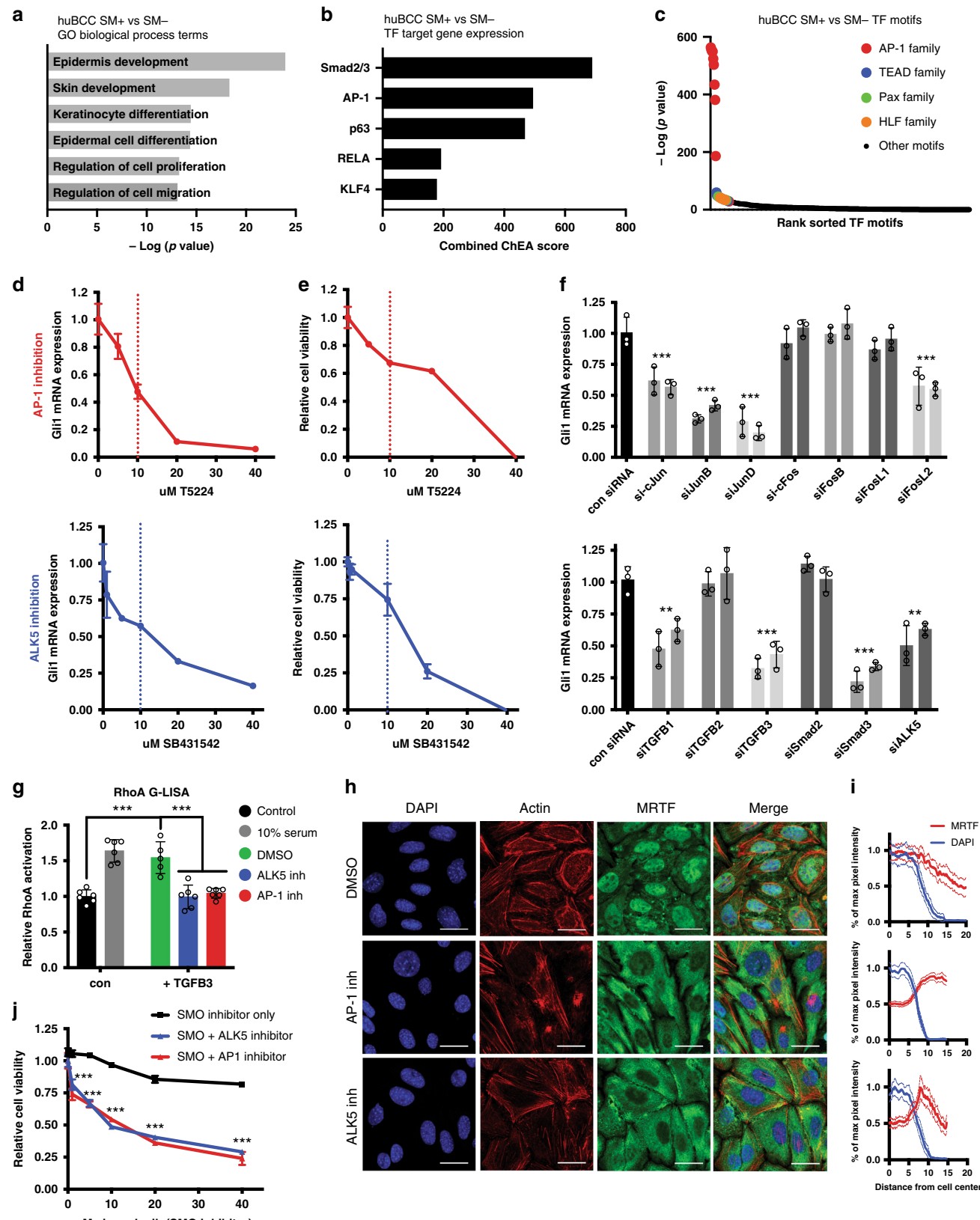

epistatic studies using small-molecule inhibitors or siRNA knock-downs at various stages of the pathway (Fig. 4i) simultaneously paired with overexpression constructs, measuring *Gli1* expression as the final output (Fig. 4j, Supplementary Fig. 4d, e). Treatment with SMO inhibitor vismodegib leads to a decrease in Hh signaling, which can be rescued by concurrent administration of TGFß3 ligand or transient overexpression of *JunD, Arhgef17, RhoA* or constitutively active *MRTF (MRTF-N)* (Fig. 4j). Importantly, the

**Fig. 3 Coincident AP-1 and TGFß signaling are required for BCC resistance. a** Top Gene Onotology (GO) Biological Process terms enriched in upregulated genes of SM+ vs. SM− huBCC cells by Binding and Expression Target Analysis (BETA) integration of RNA-seq and ATAC-seq. *p*-values calculated using Fisher exact test. **b** Transcription factors with highest combined ChIP-X Enrichment Analysis (ChEA) score enriched in SM+ vs. SM− huBCC cells by BETA integration of RNA-seq and ATAC-seq. **c** Transcription factor motifs enriched in differentially open chromatin peaks in SM+ vs. SM− huBCC cells by BETA integration of RNA-seq and ATAC-seq. **d** *Gli1* qRT-PCR in resistant BCC cell line ASZ001 treated with T5224 (AP-1 inhibitor) or SB431542 (ALK5 inhibitor) for 24 h, normalized to DMSO control. Vertical dotted line represents published IC50 in culture. **e** Cell viability of resistant BCC cell line ASZ001 treated with inhibitors for 72 h, measured by MTS assay. **f** *Gli1* qRT-PCR of resistant BCC cell line ASZ001 transfected with siRNAs against AP-1 and TGFß family genes for 48 h, compared to universal negative control siRNA. Each pair of matching-colored bars represents two distinct siRNA oligos per target gene. **p < 0.01, ***p < 0.001. **g** Rho G-LISA assay with negative control cells serum starved for 24 h, and positive control cells stimulated with 10% FBS for 5 min. ***p < 0.001. Open circles on all bar graphs represent independent biological replicates. **h** IF staining of resistant BCC cell line ASZ001 treated with DMSO, 20 µM T-5224 (AP-1 inhibitor), or 10 µM SB431542 (ALK5 inhibitor). Scale bar = 25 µm. **i** Quantification of MRTF fluorescence intensity across cell radius in (**h**), measured as µm distance from cell center. Nuclear boundaries represented as DAPI intensity. Mean intensities of n > 50 cells shown as solid lines, with SEM as dotted lines. **j** MTS cell viability assay of resistant BCC cell line ASZ001 treated with increasing doses of vismodegib (SMO[i]) only, or vismodegib + 10 µM SB431542 (ALK5 inhibitor) or 5 µM T5224 (AP-1 inhibitor). ***p < 0.001. All error bars represent mean +/− SD. *p*-values calculated using unpaired, two-tailed Student's *t*-test.

---

effects of ALK5 inhibition cannot be rescued by overexpression of *JunD* or other AP-1 subunits and conversely, AP-1 inhibition cannot be rescued by TGFß ligand administration (Fig. 4j, Supplementary Fig. 4e). These experiments together indicate that coincidental TGFß and AP-1 signaling are required for Arhgef17, RhoA, and MRTF activation that leads to non-canonical Hh signaling and tumor resistance.

**JunD/AP-1, but not TGFß, is sufficient to drive nMRTF.** To interrogate whether AP-1 or TGFß are sufficient to drive MRTF-mediated resistance, we used our NIH-3T3 model where over-expression of constitutively active MRTF (MRTF-N) in tandem with subthreshold Smoothened agonist (SAG) enhances *Gli* signaling[12]. 3T3 cells are a useful proxy for sensitive BCC cells in vitro because they respond to canonical Hh signaling but do not intrinsically express the non-canonical MRTF-driven resistance pathway[12]. We confirmed that the expression of *MRTF-N, Arhgef17, Arhgef18,* or *RhoA* is sufficient to amplify *Gli1* mRNA expression (Fig. 5a). Interestingly, we find that overexpression of *JunD* as well as other Jun family members is sufficient to amplify *Gli1*, whereas increased TGFß signaling alone is not sufficient (Fig. 5a, Supplementary Fig. 4f). We find that this sufficiency is operating at the transcriptional level, as *JunD* but not TGFß overactivation is sufficient to increase the transcription of selected GEFs (Fig. 5b). We confirmed that a basal level of TGFß signaling exists and responds to ligand stimulation with significant elevation in phosphorylated Smad2/3 levels (Supplementary Fig. 4g) and expression of the canonical target gene *Serpine1* (Fig. 5b).

Consistent with a primary role in driving nMRTF activity, the effects of *JunD* overexpression can be attenuated with inhibition of either ALK5 activity, *Arhgef17* expression, or MRTF activity (Fig. 5c). Overexpression of *Arhgef17* or *JunD* is sufficient to increase RhoA activation (Fig. 5d) and MRTF nuclear translocation in 3T3 cells (Fig. 5e, f, Supplementary Fig. 4i, j). Importantly, AP-1 activity appears to have little effect on canonical Hh signaling driven by SMO, as its inhibition does not affect *Gli1* expression levels in Hh-responsive 3T3 or C2C12 cells treated with SAG (Supplementary Fig. 4h). Therefore, we conclude that JunD/AP-1 is sufficient to amplify non-canonical Gli activity that depends on TGFß, RhoGEFs, RhoA, and MRTF signaling.

**AP-1 establishes Smad3 DNA binding profile of resistant BCC.** AP-1 has been shown to commission enhancers in conjunction with cell-type specific TFs[45]. Due to JunD, but not Smad3 suffi-ciency in driving the AP-1/Smad/Rho/MRTF resistance pathway, we hypothesized that JunD/AP-1 regulates the unique chromatin accessibility profile that functions with TGFß signaling in resistant

nMRTF BCC. Indeed, AP-1 binding motifs are highly enriched in the differential open chromatin of sorted SM+ naive human BCC (Fig. 3c), while few differences arose in canonical Smad3 binding motifs. This is consistent with previous findings that Smad pro-teins bind to their canonical DNA binding element (SBE) with 100-fold lower affinity than their interacting TF partners[46], and therefore need to cooperate with other TFs to influence tran-scription. To test this hypothesis, we conducted ATAC-seq and phospho-Smad3 chromatin immunoprecipitation sequencing (ChIP-seq) in resistant nMRTF BCC cells treated with AP-1 inhibitor. We found that AP-1 increases chromatin accessibility at Smad3 binding sites and modifies Smad3 DNA binding on a genome-wide level (Fig. 6a, Supplementary Fig. 5a–c, e). Inter-estingly, the intersection of genes dependent on AP-1 for maximal mRNA expression, chromatin accessibility, and Smad3 binding are responsible for interaction with the cytoskeleton and RhoGEF activity (Fig. 6b, c, Supplementary Data 3). These findings suggest that in resistant BCC, AP-1 shapes the chromatin accessibility landscape to open alternative Smad3 binding sites, allowing AP-1 and Smad3 to cooperatively induce expression of Rho regulators including GEFs.

To illustrate the AP-1/Smad3 chromatin cooperativity, we focused on accessibility changes in the regulatory regions of the *Arhgef17* locus, although we observed similar patterns in the regulatory regions of other GEF loci (Fig. 6d, e, Supplementary Fig. 5d). We identified an AP-1 dependent ATAC peak in resistant BCC cells within the first intron, which is closed in SMO[i]-sensitive BCCs[27], and showed that Smad3 binding within the ATAC peak disappears with AP-1 inhibition (Fig. 6d). To interrogate the function of this AP-1 dependent open chromatin region, we used the CRISPR-Cas9 system in resistant nMRTF BCC cells to delete the region of the ATAC peak (Supplementary Fig. 5f). Care was taken to ensure the neighboring exon was excluded from the targeted region (Fig. 6d) to avoid disrupting protein function. Arhgef17 ATAC-peak deletion (Arhgef17[AD]) cells displayed diminished expression of *Arhgef17* (Fig. 6f), and significant reduction of *Gli1* levels, which could be rescued by overexpression of full-length *Arhgef17, RhoA,* or *MRTF-N,* but not TGFß3 ligand stimulation (Fig. 6g, h). Arhgef17[AD] cells revealed diminished actin polymerization and decreased levels of nuclear MRTF (Fig. 6j, k), similarly to *Arhgef17* siRNA knockdown (Fig. 4g, h). Furthermore, the Arhgef17[AD] cells also display enhanced sensitivity to inhibition of Gli1 expression by SMO[i] or MRTF[i] (Fig. 6i). These genetic studies show that resistance-specific, AP-1-dependent accessible chromatin in regulatory regions of *Arhgef17* and likely other Rho activator genes are crucial for optimal Smad3 binding and transcription.

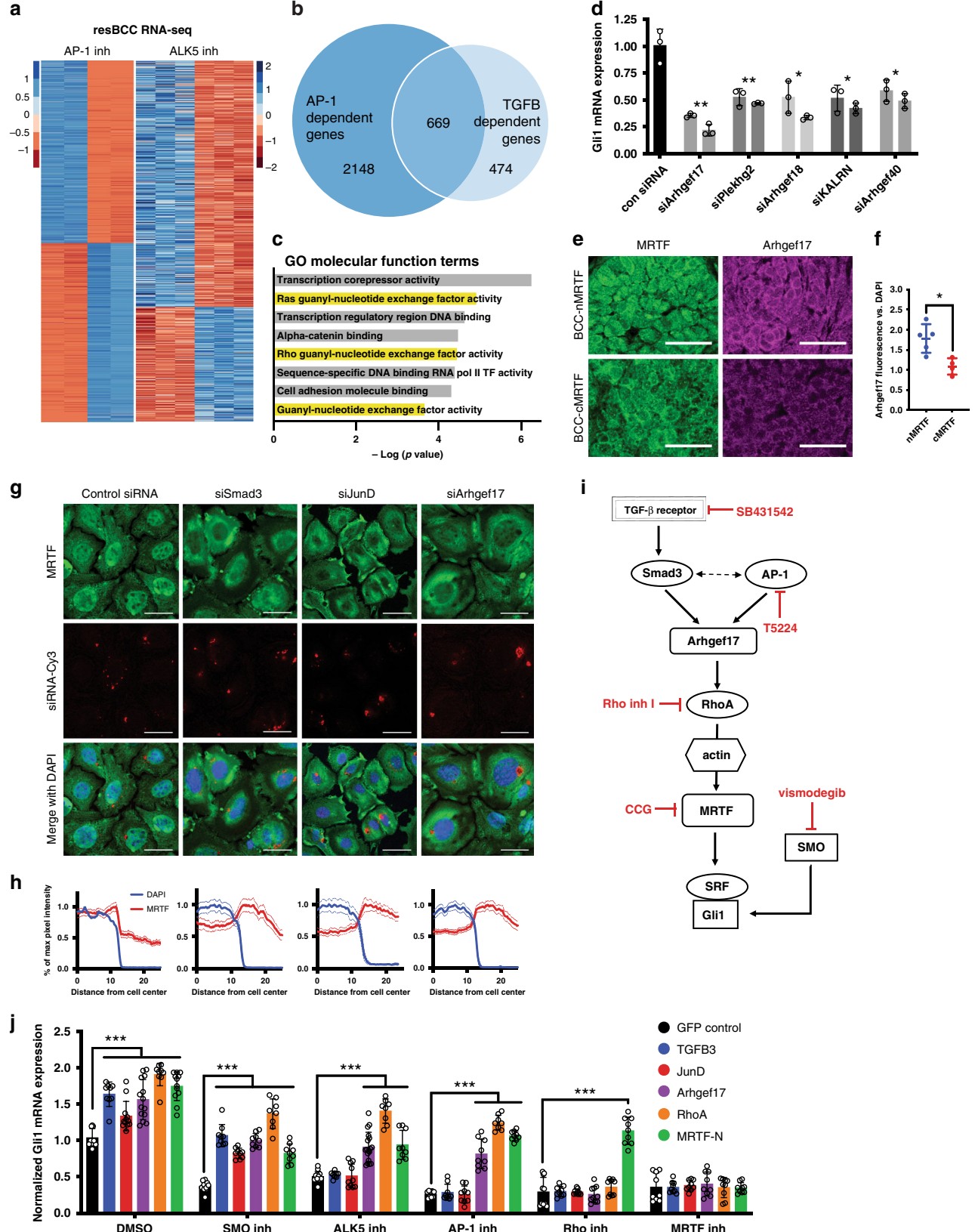

Since AP-1 signaling drives SMO[i] resistance through MRTF activation, we predicted that in treatment of patient tumors, AP-1 inhibitors, like MRTF inhibitors[12], would only display efficacy in BCC populations with nMRTF. Indeed, we see that in naive human BCC explants, tumors containing nuclear MRTF respond significantly to AP-1 inhibition with a decrease in *Gli1* expression, while tumors with only cytoplasmic MRTF are less responsive. Furthermore, there is a significant correlation between the expression of surface markers LYPD3, TACSTD2, and LY6D and the responsiveness to AP-1 inhibition (Fig. 6l). We also see

**Fig. 4 AP-1 and Smad3 induce transcription of Rho regulators including GEFs. a** Heatmap of differentially expressed genes as measured by RNA-seq of resistant ASZ001 cells treated with 10 μM SB431542 (ALK5 inhibitor) or 20 μM T5224 (AP-1 inhibitor). **b** Overlap between AP-1 and TGFß dependent genes by RNA-seq, defined as $\log_2$ FC < −1 and $p > 0.05$ in inhibitor-treated cells. Genes listed in Supplementary Data 2. **c** GO Molecular function terms enriched in genes dependent on both AP-1 and TGFß signaling. *P*-values calculated by Fisher exact test. **d** *Gli1* qRT-PCR of resistant BCC cell line ASZ001 transfected with siRNAs targeting selected GEFs for 48 h. Each pair of matching-colored bars represents two distinct siRNA oligos per target gene. *$p < 0.05$, **$p < 0.01$. **e** IF images of MRTF and Arhgef17 protein expression in murine sensitive (cMRTF) and resistant (nMRTF) BCCs[12]. Scale bar = 50 μm. **f** Quantification of (**e**) as measured by Arhgef17 fluorescence intensity vs. DAPI. Each point represents an individual tumor, mean pixel intensity normalized to mean DAPI intensity quantified over at least three 100 × 100 μm microscopy fields. *$p = 0.0107$. **g** IF images of MRTF protein localization in resistant BCC cell line ASZ001 transfected with Cy-3 conjugated siRNAs. Scale bar = 25 μm. **h** Quantification of MRTF fluorescence intensity across cell radius in (**g**), measured as μm distance from cell center. Nuclear boundaries represented as DAPI intensity. Mean intensities of $n > 50$ cells shown as solid lines, with SEM as dotted lines. **i** Diagram describing putative signaling pathway and corresponding small-molecule inhibitors. **j** Epistatic studies measured by *Gli1* qRT-PCR of resistant BCC cell line ASZ001 transfected with overexpression constructs and treated with inhibitors. ***$p < 0.001$. Open circles on all bar graphs represent independent biological replicates. All error bars represent mean +/− SD. *P*-values calculated using unpaired, two-tailed Student's *t*-test.

that patient BCC explant treatment with AP-1 inhibitors results in significantly reduced levels of nuclear MRTF (Supplementary Fig. 5g, h). We conclude that AP-1 drives the chromatin accessibility profile conducive to AP-1/Smad3-dependent nMRTF BCC resistance, and the identified surface proteins can act as prognostic markers for response rate to AP-1 inhibitors.

To further evaluate the clinical potential of these pathway inhibitors, we treated naive patient BCC explants with combinations of ALK5, AP-1, and/or SMO inhibitors. Similar to our findings in the mouse BCC cell line, SMO plus AP-1 inhibitors have an additive effect in reducing *Gli1* expression, while AP-1 plus ALK5 inhibitors do not (Supplementary Fig. 5i). These findings support the potential efficacy of combination therapies targeting the canonical and non-canonical pathways simultaneously.

## Discussion

In this study, we have identified three surface markers in BCC— LYPD3, TACSTD2, and LY6D—that reliably define the nMRTF BCC cell state and can be used as prognostic markers for resistance to SMO[i] and responsiveness to AP-1 inhibitors. We show that this resistant cell state closely resembles the differentiation state of suprabasal TACs of the hair matrix, and that AP-1 mediated chromatin accessibility facilitates a Smad3-dependent transcriptional program of RhoGEFs that drive nuclear MRTF activity and resistance. Human BCC sensitivity to AP-1 inhibitors ex vivo correlates with nMRTF status and surface marker expression, providing pre-clinical support for AP-1 inhibition as a viable target for combination therapy along with canonical SMO inhibition for resistant BCCs.

Our discovery that MRTF is active in the matrix TACs of normal skin uncovers a novel interactor with the Hh pathway, given its ability to amplify Gli signaling. Hh ligands control developmental patterning through concentration-dependent signaling in nearly every developing metazoan body tissue. In the skin, Shh is only expressed in the lateral disc, the group of cells in the matrix directly adjacent to the DP that eventually form the IRS, but Hh-responsive genes such as *ptch1* and *gli1* are expressed in proliferative matrix cells up to the line of Auber[29,47], many cell diameters away from the ligand source. Our findings suggest that MRTF reprises its role as a signaling cascade amplifier to increase the speed and distance of the paracrine Hh signal. This is consistent with our observation that MRTF inhibition during anagen severely limits the proliferative capacity of matrix cells. The observation that matrix TACs express a significant number of MRTF target genes besides those involved in Hh signaling suggests that MRTF may also play another as-yet uncharacterized developmental role in tissue patterning.

Along with the shared function of MRTF as a Hh signal amplifier, resistant BCCs and TACs also occupy similar cell states

of differentiation, as evidenced by chromatin accessibility and transcriptomes. It is important to emphasize that the data do not support that resistant BCCs are derived from matrix TACs, but rather only resemble their MRTF-dependent chromatin accessibility profile. The finding that PMXs also have active MRTF is consistent with the fact that they are derived from matrix cells, and therefore likely to maintain some elements of that chromatin accessibility as well.

The expression of the three resistance-associated surface markers, LYPD3, TACST2, and LY6D, in other contexts provide further clues of the lineage trajectory of the nMRTF cell state. LYPD3 (also known as C4.4A) is specifically expressed in suprabasal keratinocytes and is strongly associated with metastasis, invasion, and EMT[48,49]. TACSTD2 was first found in trophoblasts and other various epithelial tissues[50], while LY6D is highly related to murine hematopoietic stem cell marker Sca-1, and is a marker for early B-cell progenitors in mice[51], as well as being expressed in esophagus and interfollicular skin[52]. All three markers define specific differentiation states, and act as prognostic biomarkers for multiple solid cancers including squamous cell, colorectal, hepatocellular, pulmonary, brain, ovarian, gastric, breast, bladder, and prostate carcinoma[52].

Identification of the surface markers allowed, for the first time, prospective isolation of the nMRTF resistant cell population and defined the role for AP-1 and TGFß in maintaining the BCC resistance state. Notably, we find that MRTF activation is independent of Fos family members, with the possible exception of FosL2, suggesting that Jun proteins mediate resistance. This is further supported by the necessity of JNK signaling rather than p38/ERK1/2[17]. In the epidermis, c-Jun is expressed solely in the stratum granulosum, while JunB and JunD are expressed in the stratum basale and spinosum, indicating that they may be governing different steps in the differentiation program[16]. We have shown that JunD has the most potent effect on the TGFß/Rho/ MRTF signaling axis, but cannot rule out partial contributions from other Jun proteins.

Interaction between AP-1 and Smad TFs is also complex, having either antagonistic or cooperative functions depending on the regulatory region. Previous studies have shown that at promoters with consensus Smad binding elements (SBEs), Jun TFs inhibit Smad-dependent transcription[53], while at promoters with canonical AP-1 binding sites, or TPA response elements (TREs), Smad binding acts synergistically with Jun to enhance transcription[54,55]. This suggests that when AP-1 and Smad are simultaneously active, AP-1 shifts the Smad binding profile away from canonical SBEs and instead favors TRE activation. This is consistent with our Smad3 ChIP-seq data in resistant BCC cells, where we observe that when AP-1 is active, Smad3 binding sites are more enriched for AP-1 motifs rather than Smad motifs

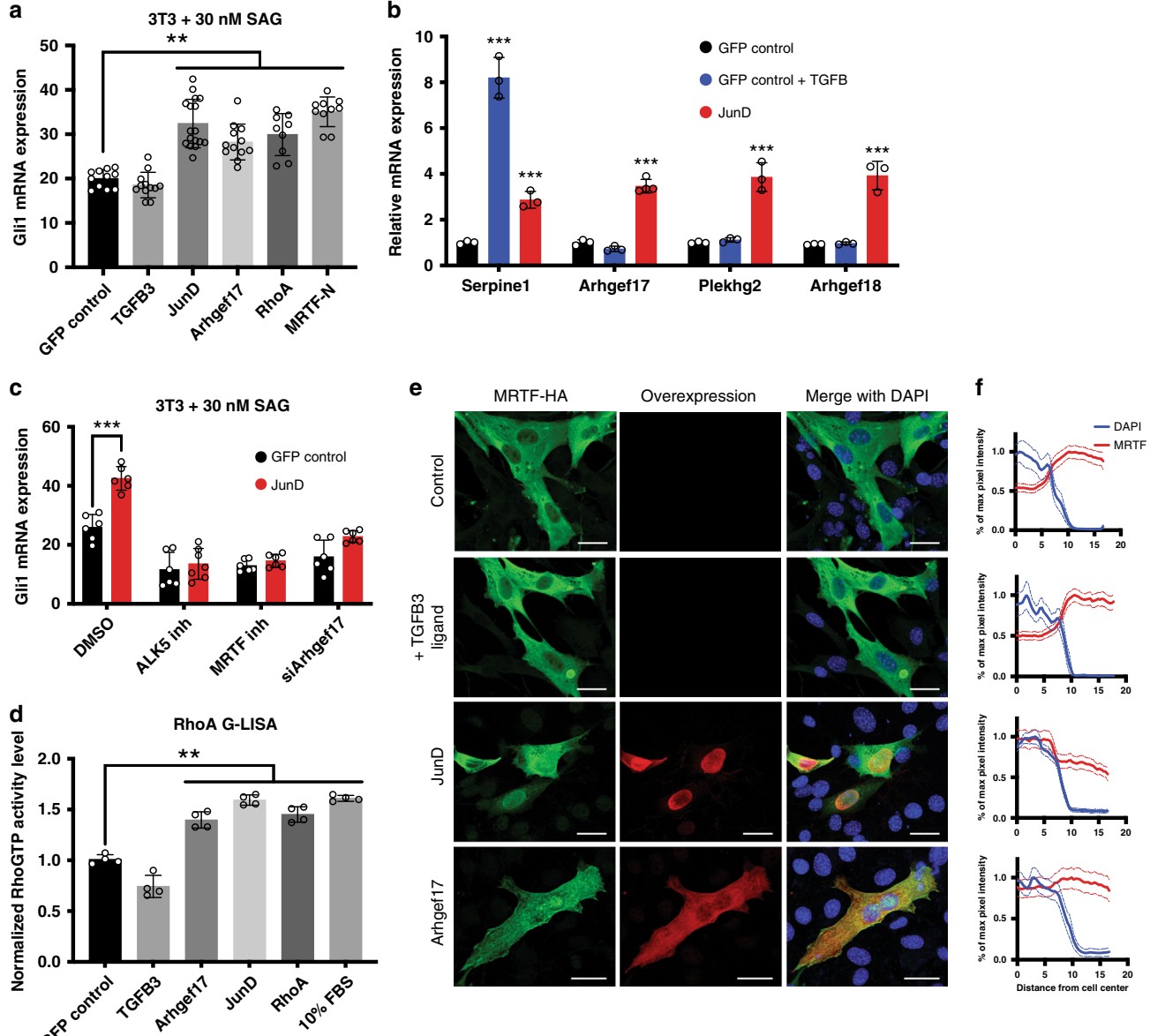

**Fig. 5 JunD/AP-1, but not TGFß, is sufficient to drive nMRTF. a** Enhancement of Hh signaling in NIH-3T3 cells measured by *Gli1* qRT-PCR transiently transfected with overexpression constructs for 48 h and treated with 30 μM Smoothened agonist (SAG) for 24 h. **p < 0.01. **b** qRT-PCR of various target genes in 3T3s transiently transfected with GFP control vectorwith or without 5 ng/ml TGFß ligand supplementation, or *JunD* overexpression construct. ***p < 0.001. **c** *Gli1* qRT-PCR in 3T3 cells transiently transfected with GFP control vector or *JunD* and treated with 30 μM Smoothened agonist (SAG) for 24 h. ***p < 0.0001. **d** RhoA activation quantified by G-LISA assay in 3T3 cells transiently transfected with GFP control or overexpression constructs for 48 h. **p < 0.01. Open circles on all bar graphs represent independent biological replicates. All error bars represent mean +/− SD. *P*-values calculated using unpaired, two-tailed Student's *t*-test. **e** IF images of 3T3 cells transfected with HA-tagged *MRTF* construct with or without 5 ng/ml recombinant TGFß ligand supplementation, *Arhgef17* or *JunD* overexpression construct. Scale bar = 25 μm. **f** Quantification of MRTF intensity across cell radius in (**e**), measured as μm distance from cell center. Nuclear boundaries represented as DAPI intensity. Mean intensities of n > 50 cells shown as solid lines, with SEM as dotted lines.

(Supplementary Fig. 5e). Whether Jun and Smad proteins are actually forming heterocomplexes on AP-1 binding sites, or binding distinct nearby cis elements is a question that is still under debate and merits further study[56–58].

Finally and importantly, we have shown that nMRTF BCC cells exist as a distinct cell state, which can be effectively targeted with small-molecule MRTF and AP-1 inhibitors, using LYPD3, TACSTD2, and LY6D as prognostic markers. We have also shown that in heterogenous tumors, combining AP-1 and SMO inhibitors can be a useful way to increase efficacy and widen the therapeutic window. Further pre-clinical in vivo studies are

necessary to gauge its potential as a therapeutic for BCC. By discovering the surface markers as a predictor of the appropriateness of combination therapies, we have advanced the possibility to target both the canonical and non-canonical drivers of resistance at once, and therefore maintain tumor therapeutic response.

## Methods

**Human samples**. Written informed consent was obtained for all human subject samples and was reviewed by the Stanford University Institutional Review Board, protocol #18325 (Stanford, CA).

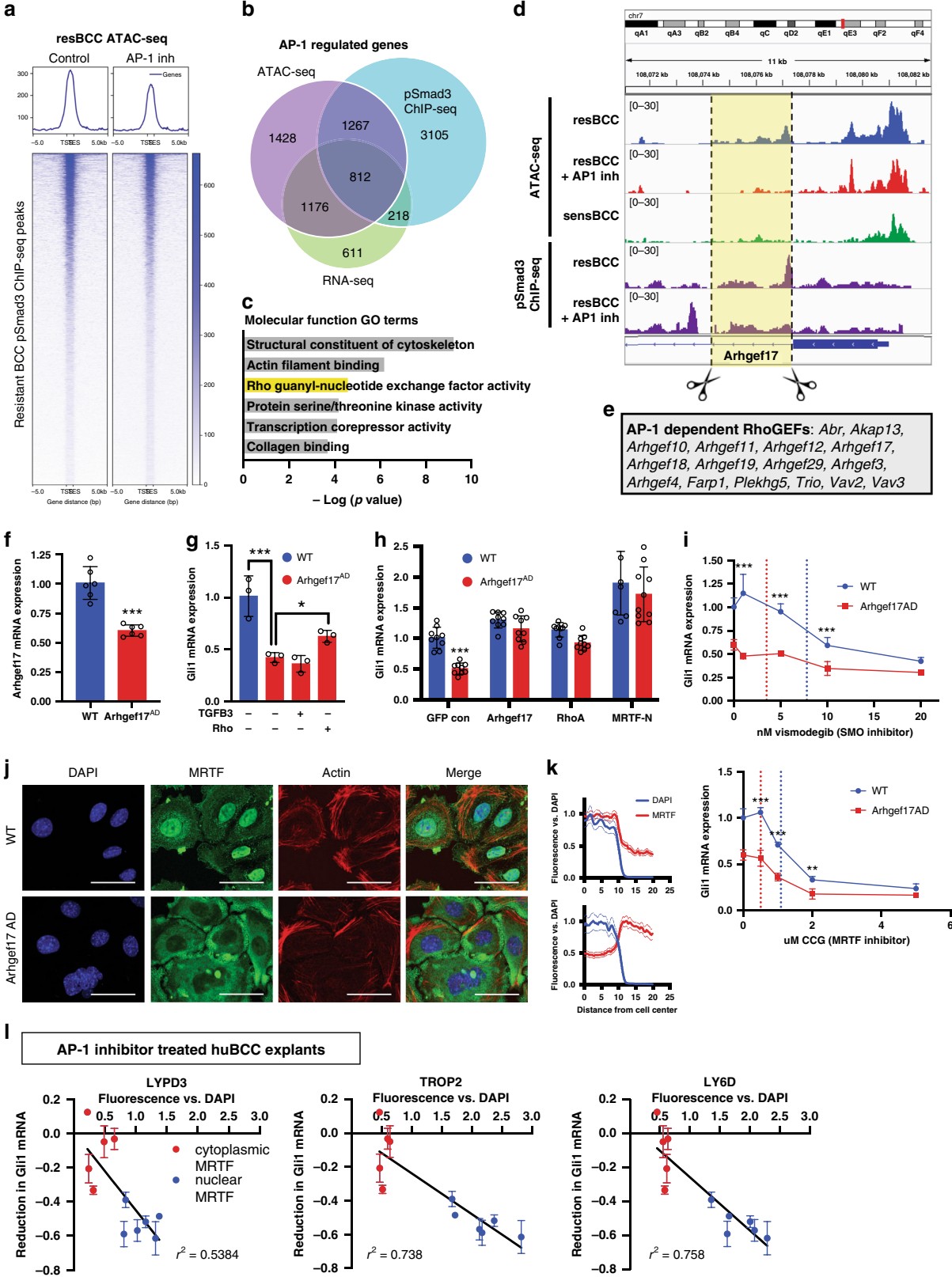

## Mice
C57BL/6 mice (Stock number: 000664) were obtained form the Jackson Laboratory. Tumor blocks for *Ptch1*[+/−];*K14-creER*;*p53* [fl/fl] mice came from previous work[12]. All mouse work was approved by the Institutional Animal Care and Use Committee (IACUC) at Stanford University.

## Mouse skin explants
C57BL/6J mice were sacrificed at p26, backs shaved, skin removed, and then 6-mm punch (Integra; 33-36) was used to generate circular punches, which were subsequently attached to the bottom of 6-well TC plates (Corning; 3516) with 6uL of Matrigel (Corning; 356237). Explants were cultured in 154CF media (Gibco; M15CF500) supplemented with 15% chelated FBS (GE HyClone; SH30396.03), 1% Penicillin Streptomycin (Gibco; 15140-122), 0.05 mM CaCl2 (Gibco; 50-9702), 0.4% Amphotericin B (Gibco; 15290026), 0.01% Plasmocin (Invitrogen; NC9886956), and 1% HKGS (Gibco; S-0001-5). Paired explants from same mouse were either treated with 150 μm CCG-1423 (Tocris; 5277) or

**Fig. 6 AP-1 establishes the Smad3 DNA binding profile of resistant BCC. a** Heatmap and line graphs of ATAC-seq signal in resistant ASZ001 cells treated with and without 20 μM AP-1 inhibitor T5224 across pSmad3 ChIP binding sites. **b** Overlap of genes displaying loss of chromatin accessibility (ATAC-seq), Smad3 binding (ChIP-seq), and/or expression (RNA-seq) in response to AP-1 inhibitor treatment in resistant BCC cells, defined as $\log_2$ FC < −1 and $p$ > 0.05. Genes listed in Supplementary Data 3. **c** GO molecular function terms enriched in genes showing AP-1 dependence of chromatin accessibility, Smad3 binding, and mRNA expression levels. *P*-values calculated by Fisher exact test. **d** Visualization of ATAC and ChIP peaks at *Arhgef17* regulatory locus. ATAC peak of interest has been highlighted, and scissors represent target sites for CRISPR guide RNAs. **e** List of RhoGEFs dependent on AP-1 by chromatin accessibility, Smad3 binding, and mRNA expression levels in resistant BCC cells. **f** *Arhgef17* qRT-PCR in WT ASZ cells and Arhgef17 ATAC-peak deletion (Arhgef17[AD]) cell line. ***$p$ < 0.0001. **g** *Gli1* qRT-PCR in WT and Arhgef17[AD] ASZ cell lines, treated with 5 ng/ml of TGFß3 ligand for 24 h or 1 μg/ml of Rho activator II for 6 h. *$p$ = 0.0344, ***$p$ = 0.0003. **h** *Gli1* qRT-PCR in WT and Arhgef17[AD] ASZ cell lines transiently transfected with overexpression constructs. ***$p$ = 0.0002. Open circles on all bar graphs represent independent biological replicates. All error bars represent mean +/− SD. *P*-values calculated using unpaired, two-tailed Student's *t*-test. **i** *Gli1* qRT-PCR in WT and Arhgef17[AD] ASZ cell lines, treated with increasing dosages of vismodegib or CCG-1423. Calculated IC50 of ASZ WT shown in blue, ARHGEF17[AD] in red. **$p$ < 0.01, ***$p$ < 0.001. **j** IF images of MRTF protein localization and actin polymerization in WT ASZ and Arhgef17[AD] cell lines. **k** Quantification of MRTF fluorescence intensity across cell radius in (**j**), measured as μm distance from cell center. Nuclear boundaries represented as DAPI intensity. Mean intensities of $n$ > 50 cells shown as solid lines, with SEM as dotted lines. **l** Correlation of relative reduction in *Gli1* mRNA expression of human tumor explants treated with 40 μM AP-1 inhibitor T-5224 for 24 h, with their relative intensity of surface marker immunostaining. Tumors were categorized as nuclear MRTF if immunofluorescent staining of MRTF colocalizing with DAPI was present in at least one of four separate 200 μm × 200 μm microscopy fields, otherwise they were categorized as cytoplasmic MRTF.

DMSO control. Explants were cultured for 72 h, then fixed in 4% PFA for 24 h and imbedded in paraffin blocks for H&E and immunofluorescence staining.

**Immunofluorescence staining and imaging**. Human and mouse BCC tumors, benign skin tumors, or normal mouse skin were fixed with 4% paraformaldehyde and embedded in paraffin blocks. 5-μm sections were mounted onto glass slides and stained with H&E or were immunolabeled using the antibodies listed below. Antigen retrieval was performed in pH 6.0 citrate buffer (Vector Laboratories H-3300). Cell lines were plated in 8-well chamber slides (Millipore) were fixed using 3.7% formaldehyde diluted in PBS for 10 min. Sections or chamber slides were immunostained using Cell Signaling Technologies IF General Protocol using the following antibodies and dilutions at 4 °C overnight: anti-Krt14 (1:500, BioLegend SIG-3476-100), anti-MKL1/MRTF (1:200, Novus NBP1-88498), anti-HHIP (1:100, Santa Cruz Technologies sc-293265), anti Krt15 (1:500, Origene BP5077), anti phospho-Smad2/3 (1:200, Abcam ab52903), anti-Ki-67 (1:500, Invitrogen MA5-14520), anti-LYPD3 (1:200, R&D Systems AF5567), anti-TACSTD2/Trop2 (1:200, abcam ab214488), anti-LY6D (1:200, Proteintech 17361-1-AP), anti-Arhgef17 (1:200, Lifespan Biosciences LS-C385259), anti-phosphoJNK (1:200, Invitrogen PA1-9594), and anti-HA tag (1:200, abcam ab130275). The fluorescent-labeled secondary antibodies used were as follows: anti–goat Alexa Fluor 488 (1:500, Life Technologies, A-11055), anti-mouse Alexa Fluor 488 (1:500, Life Technologies, A-21202), anti-rabbit Alexa Fluor 555 (1:500, Life Technologies, A-31572), anti-mouse Alexa Fluor 594 (1:500, Life Technologies, A-21203), anti-guinea pig Alexa Fluor 594 (1:500, Life Technologies A-11076), and anti-chicken Alexa Fluor 647 (1:500, Jackson Immuno Research, 703-606-155). Nuclei were visualized using Hoechst 33342 (1:1000).

Confocal imaging was carried out using a Leica SP8 microscope equipped with an adjustable white light laser and hybrid detectors, and when making comparisons between images, laser intensities and imaging settings were held constant. To quantify protein expression in BCCs, mean pixel intensity was measured using ImageJ in regions that stained positive for keratin 14, then normalized against mean pixel intensity of DAPI in the same region. For each condition, at least three $100 \times 100$ μm microscopy fields were counted in at least four tumors. Subcellular localization of MRTF was quantified through multiposition intensity profiles using the ImageJ multi-plot plug in. Line averaging was calculated on at least 50 unique nuclei per condition with the line starting in the nucleoplasm, ending in the cytoplasm, and centering on the edge of the DAPI signal. DAPI staining was used as a guide to determine nuclear localization. Measurements were then normalized to the maximum pixel intensity in each line dataset. Actin filament staining was carried out using phalloidin 488 and 647 (Life Technologies).

**Human BCC single-cell RNA-sequencing processing and analysis**. The FASTQ files for four human BCCs (BCC_2, BCC_3, BCC_8, BCC_10) were aligned using the 10X Genomics Cell Ranger 2.1.1. Each library was aligned to an indexed GRch38 genome using the Cell Ranger Count function. The Cell Ranger Aggr function was utilized to normalize the number of mapped reads across all of the sample libraries, resulting in 45,656 cells total, with median of 2,473 genes per cell and 7,935 UMI counts per cell. The clustering of cells was performed using the Seurat v2.3.4 R package[59]. In brief, initial bulk samples containing single cell data matrices were column-normalized and log-transformed. Quality control parameters were used to filter cells with 200–5000 genes with a mitochondrial percentage under 10%. Effects of cell cycle phase were calculated and regressed out using the CellCycleScoring function. The four samples were merged and then aligned using the MultiCCA function, using the top 1000 genes that were highly variable in at least two datasets. To identify cell clusters, the top 15 aligned CCs

with a resolution = 0.6 were used to obtain 22 clusters total. To present high dimensional data in two-dimensional space, we performed t-SNE analysis using the same 15 aligned CCs. For subclustering of epithelial cells, the datasets were subset for clusters with average Krt14 expression >10. Then multi-CCA alignment and clustering was repeated as above, resulting in 12 clusters.

We assigned an MRTF enrichment score using the AddModuleScore function based on a list of MRTF target genes (Supplementary Data 1). This list was generated from the intersection of genes mapped by GREAT to SRF ChIP-seq peaks[12], and genes that are differentially expressed ($\log_2$ fold change >1 or <−1, $p$ < 0.05) in resistant ASZ001 cells with MRTF inhibitor treatment[12]. Then clusters were ranked and re-ordered from 1–12 by their mean MRTF enrichment score. The 12 clusters were separated into four groups: high MRTF, med-high, med-low, and low MRTF, and differentially expressed genes were identified between the high MRTF and low MRTF groups using the FindMarkers function with default parameters.

**FACS analysis of naive human BCC tumors**. Tumors were minced, then dissociated in 0.5% collagenase in HBSS for up to 1 h, and 0.05% trypsin for up to 15 min. Cells were strained through a 70 um filter, then washed twice using FACS buffer (2% BSA /PBS), then stained with the following antibodies for 30 min at 4 °C at 1:100 dilution: anti-LYPD3 (Sino Biological 11836-R213-P), anti-Trop2/TACSTD2 (R&D systems FAB650-V), anti-LY6D (E48[60], courtesy of Dr. Ruud Brakenhoff), and anti-CD49f/ItgA6 (Millipore MAB1378). Propidium iodide was used at a concentration of 1:1000 as a dead cell marker. During FACS, live epithelial cells were first isolated by positive ItgA6 expression, then separated based on surface marker expression. Because LY6D was rarely expressed, both LY6D-positive and -negative cells were included in the surface marker positive (SM+) population, resulting in final expression profiles SM+ (ItgA6+ LYPD3+TACSTD2+LY6D+/−) vs. SM- (ItgA6+LYPD3-TACSTD2-LY6D-). Sorted cells were then used for RNA-seq and ATAC-seq. $n$ = 4 biological duplicates from two tumors. All samples were stained and FACS analyzed with the same parameters.

**RNA-seq library preparation and data processing**. RNA extraction was performed using RNeasy Plus (QIAGEN) from the samples reported in this research, per manufacturer's instructions. RNA-seq from human BCC tumors used four biological samples per condition from two tumors. RNA-seq from ASZ001 cells used two or three biological replicates per condition. Library preparation, sequencing, and mapping were carried out[3] with minor modifications. The RNA-seq libraries were constructed by TruSeq Stranded mRNA Library Prep kit (Illumina) and sequenced on an Illumina Hiseq2000 sequencer. The alignment was performed using TopHat[61] with mm9 as a reference genome. Raw counts and RPKM values were called using analyzeRepeats.pl in Homer[62]. Differential gene expression analysis was performed using the DESeq2 R package[63] with cutoffs at $\log_2$ fold change >1 or <−1 and $p$ value < 0.05. GO and ChEA enrichment analysis was performed using Enrichr[32]. Gene set enrichment analysis (GSEA)[64] was performed to determine whether a priori defined set of genes shows statistically significant difference between biological samples. The MRTF signature gene list (Supplementary Data 1) was uploaded as a custom gene set, and gene expression values from the RNA-seq analysis of the human BCC samples SM+ and SM− were used for comparison. GSEA (Broad Institute) was performed following the developer's protocol. The normalized enrichment score and nominal p value are presented to determine the significance of enrichment level.

**Cell culture**. ASZ001 and BSZ2 BCC cells were cultured in 154CF medium (Life Technologies) supplemented with 2% chelated FBS and 0.05 mM CaCl$_2$. Experiments were carried out using low serum conditions with 154CF medium containing 0.2% chelated FBS and 0.05 mM CaCl$_2$. NIH-3T3, HaCaT, and C2C12 cells (all from ATCC) were cultured in DMEM supplemented with 10% FBS. Hh induction experiments were carried out using low-serum DMEM containing 0.5% FBS supplemented with SAG (SAG, Sigma). Transient transfection of mammalian expression vectors was performed using Fugene 6 Transfection Reagent (Promega) per manufacturer protocol. Plasmids used include: pEGFP TEM4 FL (Arhgef17, Addgene 58893), pCMV Arhgef18 (Origene MC206181), pCMV Fosl2 (Origene MR204779), Flag-JunWT-Myc (Addgene 47443), pCS2 Flag-JunB (Addgene 29687), pCMV JunD (Origene MR205123), pCDH-MRTF-N*[12], pEGFP-Rho-CA[12], and pCDH-GFP[12].

**Cell viability assays**. Cells were plated subconfluently in 96-well plates, then serum starved and treated with the following inhibitors: vismodegib (GDC-0449; SelleckChem), CCG-1423 (Cayman Chemicals), T-5224 (SelleckChem), SR11302 (Cayman Chemicals), SB431542 (SelleckChem), SP600-125 (SelleckChem), JNK-IN-8 (SelleckChem), or RHO inhibitor I (Cytoskeleton Inc). Cell viability was measured after 72-h treatment using CellTiter 96 Aqueous One Solution (Promega) and normalized to DMSO vehicle control. Each condition was tested six times. For each drug and cell line, the half-maximal inhibitory concentration was determined as the drug concentration giving the half-maximal response compared with the control (DMSO-treated) conditions.

**Quantitative reverse transcription PCR**. RNA isolated using QIAGEN RNeasy Mini Kit according to manufacturer protocol. Brilliant II SYBR Green QRT-PCR 1-Step Master Mix (Agilent) was used for reverse transcription and quantitative PCR. Primer sequences are listed in Supplementary Table 2. Each reaction was performed in triplicate and values were normalized to beta-tubulin.

**Small interfering RNA (siRNA)-mediated gene knockdown**. ASZ001 cells were transfected with 30 pmol of siRNAs (Sigma-Aldrich) using Lipofectamine RNAi-MAX Reagent (Invitrogen) with standard protocol. The siRNA oligonucleotides were designed, synthesized and fluorescent labeled by Sigma-Aldrich MISSION siRNA (Supplementary Table 3). To achieve higher knockdown efficiency and avoid off-target effect, we initially screened three individual siRNAs targeting different regions and chose the top two for the subsequent study. The transfection experiment was performed over three times, and the knockdown efficiency was measured by qRT-PCR. Only the oligos that achieved successful target gene knockdown >80% were used for additional gene expression analysis. Gene expression was normalized to cells transfected with Cy-3 tagged universal negative control siRNA (Sigma-Aldrich SIC003).

**Rho activity assays**. Rho activity in ASZ001 or NIH-3T3 cells was quantified using the RhoA G-LISA colorimetric assays (Cytoskeleton Inc) per manufacturer's instructions. ASZ001 cells were grown in 6-well plates, then serum starved for 24 h and treated with 5 ng/ml TGFß3 (Peprotech), 10 µM ALK5 inhibitor SB431542 (SelleckChem), and/or 20 µM AP-1 inhibitor T-5224 (SelleckChem). NIH-3T3 cells were transiently transfected with vectors expressing GFP, Arhgef17, JunD, or RhoA for 24 h, then serum starved for 24 h. Stimulation with 10% FBS for 5 min was used as a positive control.

**Immunoblotting**. Whole-cell extracts were harvested using radio-immunoprecipitation (RIPA) buffer supplemented with protease and phosphatase inhibitors (Roche), run on SDS–PAGE gels (Life Technologies), and then transferred onto nitrocellulose membranes (0.45 µm, BioRad). Immunoblotting was carried out using antibodies against the following proteins in 5% bovine serum albumin in TBST at 4 °C overnight: GLI1 (Cell Signaling, 2534), β-tubulin (Developmental Studies Hybridoma Bank, E7), phospho-Smad2/3 (Abcam ab52903), Smad2 (BD Biosciences 610842), phospho-p38 MAPK (Cell Signaling, 4511), phospho-Smad1/5/8 (Cell Signaling, 9511). Then membranes were washed and incubated with secondary antibodies IRDye 680LT donkey anti-mouse IgG and IRDye 800CW IgG donkey anti-rabbit IgG (LI-COR). All blots were imaged using the LI-COR Odyssey image system and ImageStudioLite software.

**ATAC-Seq preparation, sequencing, processing, and analysis**. The ATAC-seq was performed[65] for sorted patient BCC cells in single cell suspension, or with minor modifications for on-plate transposition in ASZ001 cells. In brief, 100,000 ASZ001 cells were plated per replicate in a 48-well plate, then serum starved with or without 20 µM AP-1 inhibitor T-5224 for 24 h. Wells were washed with cold PBS and lysed using 0.1% NP40 in resuspension buffer for 10 min at 25 °C. Wells were washed again, then 100 µl of transposition mixture was added per well and incubated for 37 °C for 30 min, shaking at 1000 RPM using a thermoshaker. DNA was purified using QIAGEN MinElute columns and then amplified for 8–11 cycles to produce libraries for sequencing. The libraries were initially sequenced on an Illumina MiSeq sequencer and analyzed using a custom script to determine the enrichment score by calculating the ratio of signal over background at TSS over a

2-kb window. Only libraries that had the highest score above the threshold (>5) were chosen for deeper sequencing. Two independent, biological replicates were sequenced per condition.

ATAC-seq libraries were sequenced on Illumina NextSeq sequencers. Paired-end reads were trimmed for Illumina adaptor sequences and transposase sequences using a customized script and mapped to hg19 or mm9 using Bowtie1.1.2[66] with parameters –S –X2000 –m1. Duplicate reads were discarded with Samtools v0.1.18. Narrow peaks were called using MACS2[67] with parameters --nomodel --extsize 200 --shift 104 and FDR threshold 0.05. Background removal was carried out via submitting replicates to irreproducible discovery rate (IDR) filtering[68].

Overlapping peaks from all samples were merged into a unique peak list, and raw read counts mapped to each peak (tools multicov) for each individual sample were quantified. The DEseq2[63] R package was used to determine differentially accessible peaks from the merged union peak list. Cutoffs were set at log$_2$ fold change >1 or <−1 and $p < 0.05$. The heatmap and histogram were generated using the annotatePeaks.pl script in the Homer suite[69]. Heatmap data were visualized using Java TreeView. Read pileups at genomic loci were imaged using Integrated Genomics Viewer (Broad Institute). High-confidence peaks were annotated for gene associations using the Genomic Regions Enrichment of Annotations Tool (GREAT)[70]. For PCA, the union peak list was generated from ATAC-seq of ASZ001 cells as well as re-analyzed ATAC-seq data from the previously published studies listed in Supplementary Table 1. The raw counts in the union peak list were first quantile-normalized, then the PCA plot was visualized in R using prcomp.

For differential peak analysis between mouse resistant, sensitive, and residual BCC, pairwise comparisons were done using DEseq2, then peaks with significantly changing signal were selected by filtering based on threefold change and FDR 0.01. The read counts of the filtered peaks were normalized using quantile normalization and the heatmap was generated by clustering peaks and samples using the pheatmap R package.

**BETA integration of BCC ATAC-seq and RNA-seq datasets**. BETA analysis[31] was conducted using differentially upregulated peaks from human BCC (huBCC) SM+ vs. SM- ATAC-seq (log2FC > 1, adjusted $p$-value < 0.05) as the BED file input, and complete differential expression data generated by DESeq2 from huBCC SM+ vs. SM- RNA-seq as the differential gene expression input. Analysis was run on the Galaxy Cistrome platform using default parameters except for significance cutoff $p < 0.05$.

**Chromatin immunoprecipitation**. Protein–DNA complexes were isolated from ASZ001 cells, and these were used to map chromatin occupancy of phosphorylated Smad2/3[71]. ASZ001 cells were grown to 80% confluency, then serum starved for 24 h with DMSO vehicle control or 20 µM AP-1 inhibitor T5224. Cells were stimulated with 5 ng/ml TGFß3 for 2 h prior to fixation. Fifty million cells per replicate were cross-linked with 1% formaldehyde for 10 min, followed by quenching with 0.25 M glycine for 5 min at room temperature (RT). Cells were lysed in modified RIPA buffer (50 mM Tris, 150 mM NaCl$_2$, 1% Triton X-100, 0.75% SDS, 0.5% sodium deoxycholate), which was supplemented with protease and phosphatase inhibitor cocktail (Roche). Cellular extracts were sonicated using a Covaris B208 ultrasonicator to produce chromatin fragments 100–400 bp in length. The cleared extract was incubated with 10 µg of antibody against phosphoSmad2/3 (abcam ab52903) or nonspecific rabbit IgG control antibody (Cell Signaling) overnight and precipitated using magnetic Protein A Dynabeads (Invitrogen). Beads were washed with ChIP wash buffer (100 mM Tris pH 9.0, 500 mM LiCl, 1% IGEPAL, 1% deoxycholic acid, protease inhibitor cocktail (Roche)) and protein–DNA complexes were eluted with IP elution buffer (1% SDS, 50 mM NaCOH$_3$). Cross-links were reversed by incubation with 0.2 M NaCl at 67 °C overnight while shaking at 1000 r.p.m. on a thermoshaker. RNA was digested with 0.2 µg/ml RNase A at 37 °C for 30 min, and protein was digested using proteinase K for 1 h at 37 °C. DNA was isolated using Qiagen MinElute columns following the manufacturer's instructions, and used for ChIP-seq library prep.

**ChIP-sequencing library preparation and data processing**. ChIP–seq libraries were generated using the standard protocol for the NEBNext Ultra II DNA Library Prep Kit for Illumina (New England BioLabs). Each library was sequenced initially on a MiSeq sequencer to determine the quality. Only libraries containing higher signal-to-noise ratio were chosen for deeper sequencing. The ChIP-seq deep sequencing data were derived from two biological samples. ChIP libraries were sequenced using the Illumina NextSeq (400 M) platform. Sequencing reads were mapped to mm9 using Bowtie1.1.2[66] with parameters –best,–strata and –m 1 to allow for only one alignment. Duplicates are then removed using Samtools rmdup. Peaks were identified using MACS2[67] with input controls and an FDR threshold 0.05. Background removal was carried out via submitting replicates to irreproducible discovery rate (IDR) filtering[68]. The DEseq2[63] R package was used to determine differential enrichment at Smad3 binding sites with or without AP-1 inhibitor treatment. Cutoffs were set at log2 fold change >1 or <−1 and $p < 0.05$. The heatmap and histogram were generated using the annotatePeaks.pl script in the Homer suite[69]. Heatmap data were visualized using Java TreeView. Read pileups at genomic loci were imaged using Integrated Genomics Viewer (Broad Institute). High-confidence peaks were annotated for gene associations using the GREAT[70].

**Motif analysis of peaks from ATAC-seq and ChIP-seq**. Motif analysis on peak regions was performed using HOMER function findMotifsGenome.pl with default parameters to calculate the occurrence of a TF motif in peak regions compared to that in background regions. We used $-\log_{10}$ ($p$-value) to rank the enrichment level of TF motifs.

**ATAC-seq signal intensity around TF bound regions**. A window from $-5$ kb upstream to $+5$ kb downstream of pSmad3 ChIP-seq peak summits which overlapped with ATAC-seq peaks was taken and divided into 50 bp bins. The number of uniquely mapped and properly paired ATAC-seq tags overlapping each bin was counted, normalized by library size and log-transformed using an in-house script. Therefore, we generated a matrix with each row representing a peak region, and each column containing the normalized tag counts from a 50 bp bin in a consecutive manner within the 10 kb window. To visualize the signal intensity across all the TF bound regions, the data matrix was presented as a heatmap by Java TreeView. To get an average intensity, we took the mean of fragment count in each bin of all the binding regions and plotted the result as a line graph.

**CRISPR/Cas9-mediated targeted gene deletion/replacement**. We used CRISPR/Cas9-mediated genome editing to generate our Arhgef17 ATAC-peak deletion (Arhgef17^AD) ASZ001 line. We designed two gRNAs to target independent loci surrounding the 3290 bp genomic region that we wished to delete, from chr7: 108073710–108077000. The sequences of the gRNAs are as follows: TCCACAAGTCCCCGCCAAGG and GAACTGAGTACTCATTAGGC. A donor sequence with 600 bp arms flanking left and right of the region to be deleted was also used to promote homologous recombination at the locus. Both gRNAs and donor sequence were synthesized as 5′-phosphorylated gene blocks (IDT), and the gRNAs were incorporated into a DNA fragment with all components necessary for gRNA expression. Constructs were transfected into ASZ001 cells alongside an hCas9 expression plasmid, using Lipofectamine LTX Reagent with PLUS Reagent (Thermo Fisher Scientific,15338100) and the recommended manufacturer's protocol. After 48 h, we added puromycin to the cells to select out positive cells for gRNA expression. Multiple rounds of transfection and puromycin selection were conducted to achieve a purer population. The deleted region was verified by genomic PCR.

**Primary patient BCC explant culture and drug treatment**. Freshly resected tumors were obtained from patients with advanced BCC receiving Mohs surgery. Informed consent was obtained in writing for all patient samples and was reviewed by the Stanford University Institutional Review Board #18325. The tumor subtype was verified through the immediate histological examination of resected BCCs. Patient specimens were minced and cultured in EpiLife medium (Life Technologies) supplemented with 0.05 mM $CaCl_2$ with or without 1 μM of $SMO^i$ vismodegib, 40 μM AP-1 inhibitor T-5224, 10 μM AP-1 inhibitor SR11302, 40 μM ALK5 inhibitor, and/or 10 μM MRTF inhibitor CCG-1423 for 24 h. Drug-treated tissues were suspended in RLT buffer (Qiagen) and homogenized using 2 ml tissue lysing matrix E tubes (MP Biomedicals). RNA was isolated from tumors using the RNeasy standard protocol (Qiagen). RNA extracts were used to carry out qPCR with TaqMan probes for human Gli1 and Gapdh (Thermo Fisher). MRTF localization and surface marker expression was assessed in explant specimens through freezing samples in O.C.T. reagent and sectioning blocks for immunofluorescence analysis.

**Quantification and statistical analysis**. Statistical tests used in each experiment as well as information on replicates is indicated in Figure Legends. In general, data represent similar results from three or more independent, biological samples and cell cultures, unless otherwise described. Single-cell RNA sequencing is analyzed from four individual tumor datasets. Sorted human BCC cells for RNA-seq and ATAC-seq analysis include four biological replicates per condition from two tumors. Deep sequencing data of ASZ001 cells are from two or three biological samples per condition. For IF analyses of human BCC tumors, $n = 9$ tumors with nuclear MRTF and $n = 11$ tumors with cytoplasmic MRTF. For IF analyses of cultured ASZ001 and NIH-3T3 cell lines, $n > 50$ for each quantification. Scale bars, radial distribution, and quantification of pixel intensity was measured using FIJI software[72]. Bar and line graph results are presented as the mean with SD. Significance values were generated using GraphPad PRISM 8 and denoted by asterisks as indicated in figure legends. Unpaired student's $t$-test was used to determine the significance of differences with the annotations: *$p < 0.05$, **$p < 0.01$, and ***$p < 0.001$. Details of statistical methods for specific analysis are described in the corresponding methods sections. A normal distribution was observed for all data.

**Reporting summary**. Further information on research design is available in the Nature Research Reporting Summary linked to this article.

## Data availability
Sequencing data generated for this manuscript are available in the Gene Expression Omnibus using SuperSeries with GEO accession numbers GSE156855 and GSE141526. Previously published sequencing datasets used for analysis for this manuscript are available in the Gene Expression Omnibus using GEO accession numbers GSE100876, GSE116966, and GSE89928. Source data are provided with this paper.

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

## Acknowledgements

We thank BCC patients in the Stanford Clinic for their willingness to participate in our study, the Oro lab for helpful comments, and Dr. Ruud H. Brakenhoff of University Medical Center Amsterdam for generously providing the E48 antibody. The work is funded by NIH grants R37 5ARO54780 and R01ARO46786 (A.E.O.), R01CA237563 (S.X.A.), 1F30CA217017 (C.D.Y.), 5T32AR7422-37 and 1F32CA254434 (D.H.), K23 CA211793 (K.Y.S.) and Concern Foundation Award CF204525 (S.X.A.). K.Y.S. is a Damon Runyon Clinical Investigator supported (in part) by the Damon Runyon Cancer Research Foundation. Sequencing data was supported, in part, by NIH grant S10OD018220. Imaging data was supported, in part, by Award Number 1S10OD010580-01A1 from the National Center for Research Resources (NCRR). Its contents are solely the responsibility of the authors and do not necessarily represent the official views of the NCRR or the National Institutes of Health.

## Author contributions

C.D.Y. and A.E.O. conceived, executed, interpreted data and funded project; D.H., S.G., and G.S. acquired and interpreted data; S.X.A. and K.S. acquired single-cell sequencing data; R.J.W., S.M., and T.P. helped acquire and interpret data, S.A. and K.R. provided tissue and interpretation.

## Competing interests

The authors declare no competing interests.
