## [Peer Review File · Nature Communications]

Reviewers' Comments:

Reviewer #1:

Remarks to the Author:

The authors previously found that Smo inhibitor-resistant BCC express active nuclear myocardin-related transcription factor (MRTF) resulting in acquiring Smo-independent hedgehog target gene expression and persistent growth. Here, they identified molecular mechanism underlying nuclear localization of MRTF in BCC. scRNA seq with four naïve human BCC tumors revealed three surface markers, Lypd3, Tacstd2, and Ly6d highly expressed in MRTF-active BCC cells. Using these markers and epithelial marker, Itga6, they isolated Itga6+Lypd3+Trop2+Ly6d+/- cells. Small molecule inhibitors or siRNAs and overexpression experiments demonstrated JNK/AP-1 and Tgfb signaling induce the expression of Rho-related factors such as Arhgef17 leading to RhoA activation and subsequent actin polymerization and nuclear localization of MRTF in BCC. Mechanistically, JNK/AP-1 signaling established chromatin accessibility and facilitate pSmad3 DNA binding to induce the expression of Rho-related factors. These results suggest promising treatment strategy, combinatorial therapies of Smo/AP or Smo/Tgfb signaling inhibitors to target canonical and noncanonical Hh signaling in BCC, and the paper is thus a great candidate for the publication in this journal.

Addressing the following questions in the revision process may help improve the manuscript.

1. In Fig. 2f, TACSTD2 staining patterns in hu-nMRTF looks different from those in mu-nMRTF or HF matrix which are membrane-bounded. Is it because the antibody is not good for human?
2. In Fig. 4J, Gli1 expression in resistant BCC cells decreased in the presence of Smo inhibitor (a black bar) compared to DMSO treated group. If I understand correctly, the Gli1 expression is supposed to be maintained in resistant BCCs to acquire noncanonical Gli1 activity even with Smo inhibitor or that much reduction of the Gli1 expression in the presence of Smo inhibitor is normal even in resistant BCCs and does not affect oncogenic properties of BCCs? Gli1 expression in other conditions with overexpression constructs also decreased in the presence of Smo inhibitor compared to DMSO treated group. Is it because basal level of Gli1 expression decreased in resistant BCCs in the presence of Smo inhibitor?
3. In Fig. 5a, Gli1 expression in 3T3 cells with TGFB3 did not increase so they concluded TGFB3 overexpression is not sufficient to amplify Gli1. However, it is not consistent with Fig. 4J showing increased Gli1 expression in TGFB3 overexpressing BCCs. Is it because of different cell types (3T3 vs resistant BCCs)? In Fig. 5, they used 3T3 cells which are not epithelial cells or BCCs unlike other experiments. Explanation is needed.
4. In Fig. 5e, please add description for the "overexpression" column. Are these figures to show overexpressing molecules tagged by RFP? If so, why there is no RFP signaling in "TGFB3"? Is it because this is recombinant TGFB3 ligand treatment? Nuclear MRTF signal does not look stronger than cytoplasmic MRTF in immunostaining picture of ARHGEF17 overexpressing cells.
5. It would be great to show that combination of drugs, Smo/AP-1 or Smo/Tgfb signaling inhibitors reduces tumor formation and progression using in vivo mouse model or BCC transplantation to immunodeficient mice.
6. They used two different resistant BCC cell lines, ASZ001 and BSZ2 but they did not mention which cell line they used for each experiment.

Reviewer #2:

Remarks to the Author:

This paper by Yao et al extends the lab's previous work that showed an association of nuclear

translocation of myocardin related transcription factor (nMRTF) activation and SMO inhibitor resistant basal cell carcinoma (BCC). Here they isolate and study resistant nMRTF cells from human BCC samples, identify and elaborate on the mechanisms of the signals and transcription factors that lead to MRTF activation, thereby explaining the role of in non-canonical Hedgehog (Hh) signaling in innate BCC resistance. Specifically, they first show the resemblance of an nMRTF cell line to normal hair follicle (HF) matrix transit-amplifying cells (TACs) to suggest that nMRTF+ represents an inherent cellular state that promotes HF epithelial cells to respond to Shh signaling. Then they prospectively isolate nMRTF cells from tumor samples using identified surface markers (SM) in nMRTF+ state which is most enriched for AP-1 and Smad3 TF binding motifs and show that AP-1 and ALK5 inhibitors reduce Gli1 expression and cell viability in those resistant nMRTF BCC cells. Then Arhgef17 known to facilitate GDP/GTP exchange for RhoA is identified as the guanyl-nucleotide exchange factor (GEF) whose transcription is affected significantly by AP-1 and ALK5 inhibition and is suggested to regulate nMRTF nuclear translocation. Having shown that both AP-1 and TGF- β are necessary for the resistance phenotype, they also show that AP-1 but not TGF- β is able to induce MRTF activation. They explain the molecular mechanism of AP-1 acting with Smad3 by showing that AP-1 binding changes chromatin accessibility at specific sites to facilitate Smad3 binding to regulatory region of e.g. Arhgef17 that enhance MRTF activity. Overall, this is a comprehensive and detailed analysis that sheds new light on the mechanisms that drive BCC growth and resistance and offers new leads into potential effective treatments for the most common skin cancer.

Major comments:

1) The analogy to HF TAC matrix cells is intriguing; however, remains loosely connected to the work and how MRTF functions in this specific BCC resistant state. If TACs represent Hh active committed cells, it is a little confusing why this represents a mechanism for BCC resistance when it may promote terminal differentiation, as is also suggested by the activation of MRTF in pilomatrixomas which are benign and also show eventual exhausted growth with terminal differentiation. In addition, if PMXs are caused by constitutive Wnt/ β -catenin signaling, it is possible that Wnt/ β -catenin regulates MRTF activation, as this pathway is also highly upregulated in TACs. If the authors think it is necessary to include this data, perhaps they could suggest a mechanism that evades terminal differentiation, assess if other genes associated with normal HF TACs are expressed in nMRTF BCCs, or specifically examine the role of Wnt signaling in MRTF activation and smo-resistant BCCs.

2) The proposed model for AP1/TGF- β driving MRTF activation is intriguing and could be further solidified. Although the role of AP1/TGF- β was shown through a series of knockdown experiments, it is not clear how exactly the two signals function in MRTF specifically. For example, many experiments were conducted in prospectively isolated SM+ nMRTF resistant BCC cells. The epistatic experiments in 4j are revealing; however, focusing on only the SM+ population introduces bias and at least some of the experiments could be done in parallel with more controls, such as sensitive BCC cells, SM- cells or HF matrix TACs, to further substantiate that MRTF is the necessary signal amplifier in response to AP1/TGF- β signaling.

3) The authors show evidence for the importance of coincidental signaling of AP-1 and TGF- β in MRTF activation. Can the authors assess whether or not if there are additive effects of AP-1 inhibitors and TGF- β inhibitors in reducing Gli1 expression and cell viability compared to when those inhibitors are use individually? The finding that AP-1 but not TGF- β is sufficient to drive MRTF activation is explained by the complex interaction between AP-1 and Smad3 TF, which leads to the question of whether it is TGF- β signaling per se or Smad3 TF itself that cooperates with AP-1. It's possible that there are other factors for Smad3 that could also cooperate with AP-1 upon loss of Smad3, as AP-1 itself is sufficient to induce MRTF activation.

Minor comments:

1) The statistical significance of RT-qPCR results in 3f seems to be gravitated towards the experimental groups with lower baseline level of Gli1. Could try to normalize Gli1 expression across all groups according to internal control like housekeeping genes.

2) The authors identified markers that correlate with high nMRTF in fig 2 (SM+). Does inhibition of nMRTF in human bcc explants show a corresponding loss of these markers? Fig 6l shows that AP1 inhibition reduced Gli1 expression preferentially in SM+ tumors which corresponds with nMRTF+

classification. What was the nuclear MRTF status after AP1 inhibition when Gli1 levels were measured?

3) The connection between AP-1/TGF- β regulation in RhoA activation could be more developed. The knockdown of candidate Rho GEF targets shown to phenocopy the effects of Ap1 and smad3 KD are somewhat correlative.

4) It would be nice to see more experiments to validate on human BCC samples; however, this may be beyond the scope of this initial study that sets the groundwork for more human studies.

Reviewer #3:

Remarks to the Author:

Basal cell carcinoma is the most common human cancer. BCCs are driven by oncogenic Hh signaling and they were thought to be in principle treatable with the Smoothed inhibitor vismodegib. However, ~20% of patients resist treatment and they recur within a year of treatment. Resistance is thought to be either due to acquired resistance or pre-existing, intrinsically resistant tumor cell states. A subset of BCC cells expresses increased nMRTF activity making them vismidogen resistant. nMRTF is a transcription factor that partners with SRF in response to increased actin polymerization and thereby potentiates Gli-mediated gene transcription. How nMRTF activity is regulated in BCCs is still elusive. In this manuscript, Yao et al define the origin, cellular definition and transcriptional programs of the nMRTF-dependent, vismodegib resistant BCC subpopulation.

The authors detect nuclear MRTF in the hair follicle matrix, a TGF β -responsive cell population that expands dependent on Shh signaling during anlagen. They also detect nuclear MRTF staining in HF matrix-derived pilomatricomas but not benign tumors of known origins outside the HF-matrix. Consistent with this observation, ATAC-seq analyses suggest resistant BCCs are similar to basal and supra basal TAC populations (HF matrix), whereas sensitive BCCs are more closely related to cells of the hair Germ and Bulge. nMRTF inhibition in hair follicle explants causes reduced proliferation in the matrix, consistent with the idea that nMRTF has cell state specific functions that are linked to Shh signaling.

The authors utilize scRNA-seq in combination with SRF-ChIPseq and gene set enrichment analyses to define a signature score that defines nMRTF regulated targets, inclusive of three cell surface markers (LYPD3, TACSTD2, Ly6D). These markers along with α 6-integrin allows the authors to separate nMRTF positive and negative cells directly from BCC patient specimens. The authors perform further GO analyses suggestive of vismidogen resistant cells presenting a more differentiated state compared to vismidogen responsive cells. TF-motif analyses together with ChEA-analyses predict AP1, SMAD2/3, p63 as the top TFs in nMRTF+ vs nMRTF- BCCs and knock-down or pharmacological inhibition reduced nuclear nMRTF staining. Using knock down and pharmacological inhibitor studies the authors demonstrate convincingly Jun/Fos-12 regulate nMRTF downstream of JNK and canonical TGF β 1/TGF β 3-ALK5-Smad3 signaling pathway controls Hh independent Gli1 activation in nMRTF positive vismidogen resistant BCC cells. Differential expression analyses identify Arhgef17 as the most significant Rho-GEF in nMRTF positive BCC cells and they show its knockdown with siRNAs phenocopies Smad2, JunD, and AP-1 knock-down results. Epistasis experiments position Arhgef17 downstream of AP-1 and Smad3 signaling pathways, which are both required suggestive of a co-operative function these mechanisms in the activation of Rho-nMRTF/SRF leading to Hh-independent Gli1 activation and vismidogen resistance. Furthermore, over-expression experiments suggest JunD, Arhgef17, and RhoA but not Tgfb3 are sufficient to activate RhoA and translocate MRTF from the cytoplasm into the nucleus.

At last, the authors combine ATAC-seq, Smad3 ChIP-seq and AP-1 inhibitor studies to show increased AP-1 activity potentiates Smad3 dependent transcription. The authors identify a gene regulatory element that is occupied by Smad3 in an AP-1 dependent manner and deletion of this

site with CRISPR-Cas9 dependent gene editing confirms the requirement of this site for the AP-1/Smad3 dependent Arhgef17 expression.

This manuscript is very interesting and suitable for publication in Nature Communications, because it provides new insights into the mechanism by which Hh independent Gli1 activation causes vismidoigib resistance. The main strengths of the presented study are its clinical relevance, comprehensive experimental approaches (ATAC-seq, scRNA-seq, ChIP-seq), and elegant mechanistic studies that include gain and loss of function experiments as well as epistasis studies. The data are strong, well presented and correctly interpreted. The main weakness of the study is that most mechanistic work and pharmacological inhibitor studies have been conducted in cell culture models and it remains to be seen whether the presented responses are also seen in mouse and patient derived BCC models.

Minor suggestions to improve the paper:

The authors state in the introduction that a population with active nMRTF cells exists in ~60% treatment naive patient specimens, suggesting that they are intrinsically resistant to SMO inhibitor treatment. Next they write: "This observation suggests that the nMRTF phenotype exists as a distinct cell state in heterogeneous BCC, rather than simply a clonal expansion in response to selective pressures from inhibitor treatment." - This statement is confusing because even though the population may not present a genetically distinctive clone, it still pre-exists and is selected for by treatment due to its intrinsic treatment resistance. I suggest the authors consider rephrasing this statement.

In Fig. 5e MRTF-HA staining looks different upon Arhgef17 and JunD over-expression and it appears if JunD is more effective than Arhgef17 to facilitate the translocation of MRTF-HA into the nucleus. In addition, it appears that MRTF-HA is more clearly nuclear in cells that haven't been infected with JunD in the specific panel. These data appear weak and somewhat confusing. Is MRTF-HA expressed at equal levels in all the different conditions? The authors could add biochemical fractionation experiments to perhaps make this point more convincing. Furthermore, in Fig. 5e the authors may want to indicate with a line on the picture/cell where they measure/compare DAPI/MRTF in the line graph in 5f.

REVIEWER COMMENTS

Reviewer #1 (Remarks to the Author):

The authors previously found that Smo inhibitor-resistant BCC express active nuclear myocardin-related transcription factor (MRTF) resulting in acquiring Smo-independent hedgehog target gene expression and persistent growth. Here, they identified molecular mechanism underlying nuclear localization of MRTF in BCC. scRNA seq with four naïve human BCC tumors revealed three surface markers, *Lypd3*, *Tacstd2*, and *Ly6d* highly expressed in MRTF-active BCC cells. Using these markers and epithelial marker, *Itga6*, they isolated *Itga6+Lypd3+Trop2+Ly6d+/-* cells. Small molecule inhibitors or siRNAs and overexpression experiments demonstrated JNK/Ap-1 and Tgfb signaling induce the expression of Rho-related factors such as *Arhgef17* leading to RhoA activation and subsequent actin polymerization and nuclear localization of MRTF in BCC. Mechanistically, JNK/Ap-1 signaling established chromatin accessibility and facilitate pSmad3 DNA binding to induce the expression of Rho-related factors. These results suggest promising treatment strategy, combinatorial therapies of Smo/AP or Smo/Tgfb signaling inhibitors to target canonical and noncanonical Hh signaling in BCC, and the paper is thus a great candidate for the publication in this journal. Addressing the following questions in the revision process may help improve the manuscript.

We thank the reviewer for their strong support for the manuscript and appreciate the insightful comments below.

1. In Fig. 2f, TACSTD2 staining patterns in hu-nMRTF looks different from those in mu-nMRTF or HF matrix which are membrane-bounded. Is it because the antibody is not good for human?

We would like to thank the reviewer for pointing out this discrepancy. The human BCC tumors were originally stained with a relatively weak TACSTD2 antibody. Having found a more specific and sensitive antibody, the hu-nMRTF and hu-cMRTF images have been replaced with stains using the updated antibodies as listed in the methods, which display a cell membranous distribution.

2. In Fig. 4J, Gli1 expression in resistant BCC cells decreased in the presence of Smo inhibitor (a black bar) compared to DMSO treated group. If I understand correctly, the Gli1 expression is supposed to be maintained in resistant BCCs to acquire noncanonical Gli1 activity even with Smo inhibitor or that much reduction of the Gli1 expression in the presence of Smo inhibitor is normal even in resistant BCCs and does not affect oncogenic properties of BCCs? Gli1 expression in other conditions with overexpression constructs also decreased in the presence of Smo inhibitor compared to DMSO treated group. Is it because basal level of Gli1 expression decreased in resistant BCCs in the presence of Smo inhibitor?

While it is true that resistant BCCs maintain high levels of *Gli1* expression, we have shown in previous studies in both the resistant BCC cell line ASZ001 and resistant tumors generated from mouse model *K14-creER;Ptch1^{fl/fl};Tp53^{fl/fl}* that SMO inhibitor treatment will still reduce *Gli1* expression due to its effects on canonical Hedgehog signaling, albeit to a lesser degree than in sensitive BCCs. However, this partial reduction in Gli1 signaling does not impact cell growth. We and others have found that in order to see significant effects on BCC cell viability using canonical Hedgehog pathway inhibitors, Gli1 levels need to be almost completely ablated (Robarge et al. *Bioorg Med Chem Lett*

2010; Atwood et al, Cancer Cell 2015; Whitson et al, Nature Medicine 2018).

3. In Fig. 5a, Gli1 expression in 3T3 cells with TGFB3 did not increase so they concluded TGFB overexpression is not sufficient to amplify Gli1. However, it is not consistent with Fig. 4J showing increased Gli1 expression in TGFB3 overexpressing BCCs. Is it because of different cell types (3T3 vs resistant BCCs)? In Fig. 5, they used 3T3 cells which are not epithelial cells or BCCs unlike other experiments. Explanation is needed.

We thank the reviewer for this clarifying question. Fig. 4I-J show that TGF β can stimulate *Gli1* expression in BCC cells using the AP-1-driven, non-canonical resistance pathway. RNA-seq of resistant BCC cells treated with ALK5 inhibitor (Fig 4a-c) shows that TGF β /Smad3 acts through transcription of Rho GEFs. In Fig. 5, we use 3T3s for sufficiency experiments because this cell line responds to canonical Hedgehog signaling but does not intrinsically express the non-canonical resistance pathway. 3T3s serve as an *in vitro* approximation of canonical Hedgehog signaling in sensitive BCC, because unfortunately a true SMO-inhibitor sensitive BCC cell line does not currently exist. Fig. 5b shows that TGFB signaling in sensitive 3T3s is not sufficient to activate transcription of Rho GEFs, therefore the entire downstream pathway is not turned on, in contrast to the resistant BCC cell line. We show in Fig. 6 that this is because AP-1 signaling is first required to open Smad3 binding sites at RhoGEF gene promoter and enhancer sites, otherwise TGFB activation is not sufficient without the appropriate chromatin accessibility. We have added additional discussion in the text to clarify this key point.

4. In Fig. 5e, please add description for the “overexpression” column. Are these figures to show overexpressing molecules tagged by RFP? If so, why there is no RFP signaling in “TGFB3”? Is it because this is recombinant TGFB3 ligand treatment? Nuclear MRTF signal does not look stronger than cytoplasmic MRTF in immunostaining picture of ARHGEF17 overexpressing cells.

We thank the reviewer for this clarifying question and confirm that the second row contains TGFB3 ligand supplementation rather than overexpression of tagged protein. The figure and legend have been edited to clarify this. Due to the constant shuttling of MRTF in and out of the nucleus in response to dynamic actin polymerization levels, cells with active MRTF will not always show stronger nuclear MRTF staining than cytoplasmic. However, the quantification over at least 50 cells in Fig 5f shows a clear net gain in nuclear MRTF in response to Arhgef17 or JunD overexpression.

5. It would be great to show that combination of drugs, Smo/AP-1 or Smo/Tgfb signaling inhibitors reduces tumor formation and progression using *in vivo* mouse model or BCC transplantation to immunodeficient mice.

While we would have liked to include additional murine *in vivo* studies in this manuscript, the unprecedented circumstances of the Covid-19 pandemic forced us to dramatically reduce our BCC mouse model colonies, preventing us from including the suggested mouse tumor experiments within a reasonable amount of time. In lieu of mouse studies, we have included new data with

combination drug treatments both in naïve patient BCC explants (Supplementary Fig. 5i) as well as in the resistant murine BCC cell line ASZ001 (Fig. 3j, Supplementary Fig. 3f), measuring reduction in Hedgehog signaling through *Gli1* expression and cell viability. As our model predicts, we find that combination therapies of SMO and AP-1 inhibitors have additive effects compared to either inhibitor alone, suggesting we are targeting separate pathways simultaneously, while there is no additive effect when combining AP-1 and TGF β /ALK5 inhibitors, suggesting that these signals are part of the same pathway.

6. They used two different resistant BCC cell lines, ASZ001 and BSZ2 but they did not mention which cell line they used for each experiment.

We thank the reviewer for highlighting this ambiguity. Descriptions have been added to figure legends. For reference, the ASZ001 cell line was used for experiments in Fig. 1c, 3e-j, 4a-d, 4g-j, and 6a-k, while the BSZ2 cell line was used in Supplementary Fig. 3i.

Reviewer #2 (Remarks to the Author):

This paper by Yao et al extends the lab's previous work that showed an association of nuclear translocation of myocardin related transcription factor (nMRTF) activation and SMO inhibitor resistant basal cell carcinoma (BCC). Here they isolate and study resistant nMRTF cells from human BCC samples, identify and elaborate on the mechanisms of the signals and transcription factors that lead to MRTF activation, thereby explaining the role of in non-canonical Hedgehog (Hh) signaling in innate BCC resistance. Specifically, they first show the resemblance of an nMRTF cell line to normal hair follicle (HF) matrix transit-amplifying cells (TACs) to suggest that nMRTF+ represents an inherent cellular state that promotes HF epithelial cells to respond to Shh signaling. Then they prospectively isolate nMRTF cells from tumor samples using identified surface markers (SM) in nMRTF+ state which is most enriched for AP-1 and Smad3 TF binding motifs and show that AP-1 and ALK5 inhibitors reduce *Gli1* expression and cell viability in those resistant nMRTF BCC cells. Then *Arhgef17* known to facilitate GDP/GTP exchange for RhoA is identified as the guanyl-nucleotide exchange factor (GEF) whose transcription is affected significantly by AP-1 and ALK5 inhibitor and is suggested to regulate nMRTF nuclear translocation. Having shown that both AP-1 and TGF- β are necessary for the resistance phenotype, they also show that AP-1 but not TGF- β is able to induce MRTF activation. They explain the molecular mechanism of AP-1 acting with Smad3 by showing that AP-1 binding changes chromatin accessibility at specific sites to facilitate Smad3 binding to regulatory region of e.g. *Arhgef17* that enhance MRTF activity. Overall, this is a comprehensive and detailed analysis that sheds new light on the mechanisms that drive BCC growth and resistance and offers new leads into potential effective treatments for the most common skin cancer.

We thank the reviewer for their strong support and have endeavored to address their insightful comments below.

Major comments:

- 1) The analogy to HF TAC matrix cells is intriguing; however, remains loosely connected to the work and how MRTF functions in this specific BCC resistant state. If TACs represent Hh active committed cells, it is a little confusing why this represents a mechanism for BCC resistance when it may promote terminal differentiation, as is also suggested by the activation of MRTF in pilomatrixomas which are benign and also show eventual exhausted growth with terminal differentiation. In addition, if PMXs are caused by constitutive Wnt/b-catenin

signaling, it is possible that Wnt/bcatenin regulates MRTF activation, as this pathway is also highly upregulated in TACs. If the authors think it is necessary to include this data, perhaps they could suggest a mechanism that evades terminal differentiation, assess if other genes associated with normal HF TACs are expressed in nMRTF BCCs, or specifically examine the role of Wnt signaling in MRTF activation and smo-resistant BCCs.

The reviewer raises several interesting points. First, we respectfully disagree with the statement that our findings in TACs are only loosely connected to BCC resistance; we believe that the connection between normal differentiation and malignancy can be very instructive. According to our ATAC-seq analysis, resistant BCCs have a similar chromatin accessibility profile to matrix TACs, while sensitive BCCs resemble bulge stem cells (Figure 1c). While TACs are more differentiated than bulge cells, they are a transient highly proliferative cell state driven by high Hh signaling, and therefore would not be considered “terminally differentiated”.

Next, we would like to emphasize that we are not claiming that resistant BCCs are actually derived from matrix TACs, we only drew the comparison in the context of their shared differentiation state and the way MRTF acts as a hedgehog signal amplifier in both cell types (Figure 1d-g). This suggests that the differentiation state establishes a specific chromatin accessibility profile that is conducive to MRTF activation. Strikingly, two-thirds of MRTF target genes identified in resistant BCCs are also expressed specifically in basal and/or suprabasal TACs (Supplementary Fig. 1c), suggesting that there may also be other MRTF-driven mechanisms shared between TACs and malignant cells that are independent of Hedgehog signaling.

The finding that PMXs also have active MRTF is consistent with the fact that they are derived from matrix cells and therefore likely maintain some elements of that chromatin accessibility. However, because PMXs are driven by constitutive Wnt/b-catenin without high Hh signaling, their phenotypes diverge from BCCs and TACs, and they are instead benign with terminal differentiation and exhausted growth. Therefore, it is more likely that MRTF and Wnt, like MRTF and Hh, have a parallel relationship rather than a causal one. Potential interaction points could form the basis of future studies. We have added additional discussion to clarify these key points in the text.

2) The proposed model for AP1/TGF- β driving MRTF activation is intriguing and could be further solidified. Although the role of AP1/TGF- β was shown through a series of knockdown experiments, it is not clear how exactly the two signals function in MRTF specifically. For example, many experiments were conducted in prospectively isolated SM+ nMRTF resistant BCC cells. The epistatic experiments in 4j are revealing; however, focusing on only the SM+ population introduces bias and at least some of the experiments could be done in parallel with more controls, such as sensitive BCC cells, SM- cells or HF matrix TACs, to further substantiate that MRTF is the necessary signal amplifier in response to AP1/TGF- β signaling.

Our proposed mechanism for how AP-1 and TGFB activate MRTF is detailed in Fig. 4i: Smad3 and AP-1 turn on transcription of regulators of Rho including GEFs, Rho activation leads to increased actin polymerization, allowing the accumulation of nuclear MRTF. We confirm that RhoGEFs are target genes of AP-1 and Smad3 by ATAC-seq, RNA-seq, and ChIP-seq analyses (Fig. 6a-e). Unfortunately, parallel perturbation experiments in sensitive BCC cells are not possible because sensitive BCCs are not amenable to culture. However, we have shown the sufficiency of AP-1 and MRTF in amplifying

Hedgehog signaling in 3T3 cells, which we use as a proxy for sensitive BCCs because they respond to canonical Hedgehog signaling and do not intrinsically express the resistance pathway (Fig. 5a-f, Supplementary Fig. 4f-j). Furthermore, we have now added additional data to substantiate that AP-1 is required for MRTF and Hedgehog activation in a clinically relevant system by treating naïve patient BCC explants with AP-1 inhibitor and seeing reductions in MRTF nuclear localization as well as Gli1 expression (Supplementary Fig. 5g-i).

3) The authors show evidence for the importance of coincidental signaling of AP-1 and TGF- β in MRTF activation. Can the authors assess whether or not if there are additive effects of AP-1 inhibitors and TGF- β inhibitors in reducing Gli1 expression and cell viability compared to when those inhibitors are used individually? The finding that AP-1 but not TGF- β is sufficient to drive MRTF activation is explained by the complex interaction between AP-1 and Smad3 TF, which leads to the question of whether it is TGF- β signaling per se or Smad3 TF itself that cooperates with AP-1. It's possible that there are other factors for Smad3 that could also cooperate with AP-1 upon loss of Smad3, as AP-1 itself is sufficient to induce MRTF activation.

We thank the reviewer for this comment and have added new combination drug treatment data in both in naïve patient BCC explants (Supplementary Fig. 5i) as well as in the resistant murine BCC cell line ASZ001 (Supplementary Fig. 3f), measuring reduction in Hedgehog signaling through *Gli1* expression and cell viability. We find that there is no additive effect when combining AP-1 and TGF β /ALK5 inhibitors, suggesting that these signals are part of the same pathway. While it is definitely possible that there are other transcription factors that cooperate with AP-1 other than Smad3, we have shown that Smad3 (and not Smad2) activity is necessary to activate the resistance pathway in the context of intact AP-1 signaling (Fig. 3f, 5c), even though it is not sufficient.

Minor comments:

1) The statistical significance of RT-qPCR results in 3f seems to be gravitated towards the experimental groups with lower baseline level of Gli1. Could try to normalize Gli1 expression across all groups according to internal control like housekeeping genes.

For all qRT-PCR experiments in this manuscript, Ct values are first normalized against housekeeping gene beta-tubulin as stated in the Methods section. For Fig. 3f, all *Gli1* expression levels are displayed in comparison to universal negative control siRNA (the first column). Each pair of matching-colored bars represents two distinct siRNA oligos used for each gene knockdown. The figure legend has been edited to clarify this.

2) The authors identified markers that correlate with high nMRTF in fig 2 (SM+). Does inhibition of nMRTF in human bcc explants show a corresponding loss of these markers? Fig 6l shows that AP1 inhibitor reduced Gli1 expression preferentially in SM+ tumors which corresponds with nMRTF+ classification. What was the nuclear MRTF status after AP1 inhibitor when Gli1 levels were measured?

We thank the reviewer for this insightful question. Indeed we now include new data showing that

human BCC explants treated with MRTF inhibitor CCG-1423 demonstrate decreased protein levels of all three surface markers (Fig. 2h), supporting our hypothesis that these genes are targets of MRTF. Consistent with our previous studies (Whitson et al, Nature Medicine 2018), we see that naïve patient BCCs display a wide range of nuclear MRTF levels (from 30-80%), but after *ex vivo* AP-1 inhibitor treatment, nuclear MRTF levels drop below 20%, indicating that MRTF activation is dependent on AP-1 signaling (Supplementary Fig. 5g-h).

3) The connection between AP-1/TGF- β regulation in RhoA activation could be more developed. The knockdown of candidate Rho GEF targets shown to phenocopy the effects of Ap1 and smad3 KD are somewhat correlative.

While it is true that the GEF knockdowns are correlative, we have also shown the direct effect of AP-1 and TGF β on RhoA activation using the Rho G-LISA assay, which directly measures changes in levels of GTP-bound RhoA following TGF β /AP-1 inhibition or overexpression in resistant or sensitive cells. As we would expect, resistant cells treated with TGF β or AP-1 inhibition showed decreased amounts of active RhoA (Fig. 3g), while sensitive cells with AP-1 overexpression were sufficient to increase levels of active RhoA (Fig. 5d, Supplementary Fig. 4f).

4) It would be nice to see more experiments to validate on human BCC samples; however, this may be beyond the scope of this initial study that sets the groundwork for more human studies.

We thank the reviewer for the comment. We have added several more experiments from patient BCC samples that validate or support our *in vitro* findings:

- a. Fig 2h: treatment of human BCC explants with MRTF inhibitor shows that expression of surface markers LYPD3, TACSTD2, and LY6D are dependent on MRTF activation.
- b. Supplementary Fig. 5g-h: treatment of human BCC explants with AP-1 inhibitor results in loss of nuclear MRTF localization.
- c. Supplementary Fig. 5i: Combination treatments of human BCC explants with ALK5 inhibitor, AP-1 inhibitor, and/or SMO inhibitor shows that SMO plus AP-1 inhibitors have additive effects and are promising candidates for dual therapy, while AP-1 and ALK5 inhibitors are likely targeting the same pathway and do not have additive effects.

Reviewer #3 (Remarks to the Author):

Basal cell carcinoma is the most common human cancer. BCCs are driven by oncogenic Hh signaling and they were thought to be in principle treatable with the Smoothed inhibitor vismodegib. However, ~20% of patients resist treatment and they recur within a year of treatment. Resistance is thought to be either due to acquired resistance or pre-existing, intrinsically resistant tumor cell states. A subset of BCC cells expresses increased nMRTF activity making them vismidogen resistant. nMRTF is a transcription factor that partners with SRF in response to increased actin polymerization and thereby potentiate Gli-mediated gene transcription. How nMRTF activity is regulated in BCCs is still elusive. In this manuscript, Yao et al define the origin, cellular definition and transcriptional programs of the nMRTF-dependent, vismodegib resistant BCC subpopulation.

The authors detect nuclear MRTF in the hair follicle matrix, a TGF β -responsive cell population that expands dependent on Shh signaling during anlagen. They also detect nuclear MRTF staining in HF matrix-derived

pilomatricomas but not benign tumors of known origins outside the HF-matrix. Consistent with this observation, ATAC-seq analyses suggest resistant BCCs are similar to basal and supra basal TAC populations (HF matrix), whereas sensitive BCCs are more closely related to cells of the hair Germ and Bulge. nMRTF inhibition in hair follicle explants causes reduced proliferation in the matrix, consistent with the idea that nMRTF has cell state specific functions that are linked to Shh signaling.

The authors utilize scRNA-seq in combination with SRF-ChIPseq and gene set enrichment analyses to define a signature score that defines nMRTF regulated targets, inclusive of three cell surface markers (LYPD3, TACSTD2, Ly6D). These markers along with $\alpha 6$ -integrin allows the authors to separate nMRTF positive and negative cells directly from BCC patient specimens. The authors perform further GO analyses suggestive of vismidogen resistant cells presenting a more differentiated state compared to vismidogen responsive cells. TF-motif analyses together with ChEA-analyses predict AP1, SMAD2/3, p63 as the top TFs in nMRTF+ vs nMRTF- BCCs and knock-down or pharmacological inhibition reduced nuclear nMRTF staining. Using knock down and pharmacological inhibitor studies the authors demonstrate convincingly Jun/Fos-l2 regulate nMRTF downstream of JNK and canonical TGF β 1/TGF β 3-ALK5-Smad3 signaling pathway controls Hh independent Gli1 activation in nMRTF positive vismidogen resistant BCC cells.

Differential expression analyses identify Arhgef17 as the most significant Rho-GEF in nMRTF positive BCC cells and they show its knockdown with siRNAs phenocopies Smad2, JunD, and AP-1 knock-down results. Epistasis experiments position Arhgef17 downstream of AP-1 and Smad3 signaling pathways, which are both required suggestive of a co-operative function these mechanisms in the activation of Rho-nMRTF/SRF leading to Hh-independent Gli1 activation and vismidogen resistance. Furthermore, over-expression experiments suggest JunD, Arhgef17, and RhoA but not Tgfb 3 are sufficient to activate RhoA and translocate MRTF from the cytoplasm into the nucleus.

At last, the authors combine ATAC-seq, Smad3 ChIP-seq and AP-1 inhibitor studies to show increased AP-1 activity potentiates Smad3 dependent transcription. The authors identify a gene regulatory element that is occupied by Smad3 in an AP-1 dependent manner and deletion of this site with CRISPR-Cas9 dependent gene editing confirms the requirement of this site for the AP-1/Smad3 dependent Arhgef17 expression.

This manuscript is very interesting and suitable for publication in Nature Communications, because it provides new insights into the mechanism by which Hh independent Gli1 activation causes vismidogib resistance. The main strengths of the presented study are its clinical relevance, comprehensive experimental approaches (ATAC-seq, scRNA-seq, ChIP-seq), and elegant mechanistic studies that include gain and loss of function experiments as well as epistasis studies. The data are strong, well presented and correctly interpreted. The main weakness of the study is that most mechanistic work and pharmacological inhibitor studies have been conducted in cell culture models and it remains to be seen whether the presented responses are also seen in mouse and patient derived BCC models.

We thank the reviewer for their strong support of our work.

Minor suggestions to improve the paper:

The authors state in the introduction that a population with active nMRTF cells exists in ~60% treatment naive

patient specimens, suggesting that they are intrinsically resistant to SMO inhibitor treatment. Next the write: "This observation suggests that the nMRTF phenotype exists as a distinct cell state in heterogeneous BCC, rather than simply a clonal expansion in response to selective pressures from inhibitor treatment." - This statement is confusing because even though the population may not present a genetically distinctive clone, it still pre-exists and is selected for by treatment due to its intrinsic treatment resistance. I suggest the authors consider rephrasing this statement.

We thank the reviewer for pointing out this ambiguity and have removed the second half of the sentence from the text for the sake of clarity.

In Fig. 5e MRTF-HA staining looks different upon Arhgef17 and JunD over-expression and it appears if JunD is more effective than Arhgef17 to facilitate the translocation of MRTF-HA into the nucleus. In addition, it appears that MRTF- HA is more clearly nuclear in cells that haven't been infected with JunD in the specific panel. These data appear weak and somewhat confusing. Is MRTF-HA expressed at equal levels in all the different conditions? The authors could add biochemical fractionation experiments to perhaps make this point more convincing. Furthermore, in Fig. 5e the authors may want to indicate with a line on the picture/cell where they measure/compare DAPI/MRTF in the line graph in 5f.

The weak nuclear signal seen in the green channel in non-transfected cells in Figure 5e appears to be a staining artifact from the HA antibody, as HA should not be expressed in the non-transfected cells. We have since repeated the staining and updated the images to show only the strong HA signal in transfected cells. While each condition was transfected with the same concentration of MRTF-HA expressing plasmid, it is difficult to completely standardize the expression levels resulting from the transfection, so there may be differences across conditions. However, each experiment was repeated as least 3 times and representative images are shown. Furthermore, due to the constant shuttling of MRTF in and out of the nucleus in response to dynamic actin polymerization levels, cells with active MRTF will not always show stronger nuclear MRTF staining than cytoplasmic, depending on when the images are taken. To account for this, the quantification Fig 5f shows the average distribution of MRTF and DAPI fluorescence over at least 50 cells per condition, and therefore we can conclude that there is a clear net gain in nuclear MRTF in response to Arhgef17 or JunD overexpression.

Reviewers' Comments:

Reviewer #1:

Remarks to the Author:

The authors addressed my comments and I do not have additional comment.

Reviewer #2:

Remarks to the Author:

Yao et al have provided additional discussion and additional experiments to this work, which have significantly strengthened the conclusions and clarified the questions/comments. Overall, this is a very important and timely study that provides new insight into the complex mechanisms of BCC heterogeneity and BCC resistance and will be a relevant reference and basis for future therapeutic efforts.

Minor comment:

The connection to normal hair follicle TACs is very intriguing and the authors provided additional clarification. It would be interesting to speculate on the signals or mechanisms that allow BCC cells to evade terminal differentiation in contrast to TACs that eventually terminally differentiate. The authors state that nMRTF BCCs are not actually derived from matrix TACs but rather resemble them. Along the same vein, BCCs are not universally thought to "derive from bulge and ORS" (1st paragraph of Results section).

Reviewer #3:

Remarks to the Author:

The revised manuscript by Yao et al. addressed my comments. However, given that Figure 5f illustrates average measurements of >50 cells, it would be useful to plot the error range along with the average value to better illustrate that the line is not only a representative but rather a statistically significant representation. Besides this minor concern - I find this manuscript exciting and of interest to the Nature communications readership.

REVIEWERS' COMMENTS:

Reviewer #1 (Remarks to the Author):

The authors addressed my comments and I do not have additional comment.

Reviewer #2 (Remarks to the Author):

Yao et al have provided additional discussion and additional experiments to this work, which have significantly strengthened the conclusions and clarified the questions/comments. Overall, this is a very important and timely study that provides new insight into the complex mechanisms of BCC heterogeneity and BCC resistance and will be a relevant reference and basis for future therapeutic efforts.

Minor comment:

The connection to normal hair follicle TACs is very intriguing and the authors provided additional clarification. It would be interesting to speculate on the signals or mechanisms that allow BCC cells to evade terminal differentiation in contrast to TACs that eventually terminally differentiate. The authors state that nMRTF BCCs are not actually derived from matrix TACs but rather resemble them. Along the same vein, BCCs are not universally thought to “derive from bulge and ORS” (1st paragraph of Results section).

RESPONSE: A clarifying statement has been added to the discussion.

Reviewer #3 (Remarks to the Author):

The revised manuscript by Yao et al. addressed my comments. However, given that Figure 5f illustrates average measurements of >50 cells, it would be useful to plot the error range along with the average value to better illustrate that the line is not only a representative but rather a statistically significant representation. Besides this minor concern - I find this manuscript exciting and of interest to the Nature communications readership.

RESPONSE: Dotted lines representing SEM have been added to all line graphs quantifying fluorescence across cell radii.